# Chemotherapy-Induced Cognitive Impairment and Hippocampal Neurogenesis: A Review of Physiological Mechanisms and Interventions

**DOI:** 10.3390/ijms222312697

**Published:** 2021-11-24

**Authors:** Melanie J. Sekeres, Meenakshie Bradley-Garcia, Alonso Martinez-Canabal, Gordon Winocur

**Affiliations:** 1School of Psychology, University of Ottawa, Ottawa, ON K1N 6N5, Canada; mbrad048@uottawa.ca; 2Cell Biology Department, National Autonomous University of Mexico, Mexico City 04510, Mexico; acanabal@ciencias.unam.mx; 3Rotman Research Institute, Baycrest Center, Toronto, ON M6A 2E1, Canada; gwinocur@research.baycrest.org; 4Department of Psychology, Department of Psychiatry, University of Toronto, Toronto, ON M5S 3G3, Canada; 5Department of Psychology, Trent University, Peterborough, ON K9J 7B8, Canada

**Keywords:** chemobrain, chemotherapy induced cognitive impairment, hippocampus, neurogenesis, dentate gyrus, memory, pre-clinical models, rodent

## Abstract

A wide range of cognitive deficits, including memory loss associated with hippocampal dysfunction, have been widely reported in cancer survivors who received chemotherapy. Changes in both white matter and gray matter volume have been observed following chemotherapy treatment, with reduced volume in the medial temporal lobe thought to be due in part to reductions in hippocampal neurogenesis. Pre-clinical rodent models confirm that common chemotherapeutic agents used to treat various forms of non-CNS cancers reduce rates of hippocampal neurogenesis and impair performance on hippocampally-mediated learning and memory tasks. We review the pre-clinical rodent literature to identify how various chemotherapeutic drugs affect hippocampal neurogenesis and induce cognitive impairment. We also review factors such as physical exercise and environmental stimulation that may protect against chemotherapy-induced neurogenic suppression and hippocampal neurotoxicity. Finally, we review pharmacological interventions that target the hippocampus and are designed to prevent or reduce the cognitive and neurotoxic side effects of chemotherapy.

## 1. Introduction

Cancer survivors frequently suffer cognitive disturbances following chemotherapy (‘chemobrain’) that include, in particular, memory loss associated with hippocampal dysfunction [1]. Initially, reports of chemotherapy-induced cognitive impairment (CICI) were largely attributed to psychological distress. However, by the early 2000s, as research into the condition increased, cognitive and neuroimaging investigations identified neurological disruptions, including white matter abnormalities in the brains of chemotherapy-treated patients [2,3,4,5,6,7]. These findings provided important evidence that CICI is related to neurotoxic alterations in the brain.

It is now widely recognized that common chemotherapeutic agents are capable of inducing post-treatment changes to overall brain volume, including persistent alterations in both white and gray matter which can lasting up to twenty years following treatment [1,8,9]. Chemotherapy-related changes in white matter integrity have been observed specifically in the prefrontal cortex and temporal lobe [10,11], regions responsible for mediating executive functioning and memory processing, with the severity of impairment increasing with higher dosage treatments [12]. Confounding factors such as age of disease onset, treatment protocols, hormonal therapy, menopausal stage, and psychosocial factors such as environmental, social, and employment-related stress all contribute to the difficulties in identifying the physiological basis of chemotherapy-related neurotoxicity in patients [13]. Pre-clinical animal models have been shown to be useful in controlling for these limitations.

Pre-clinical studies of the effects of anti-cancer drugs on rats and mice have replicated several common cognitive impairments observed in patients, including hippocampally-mediated memory impairment and working memory deficits. These findings indicate that rodent models of CICI have a high degree of prima facie validity for assessing chemotherapy-related behavioral disturbances. Critically, this allows for more controlled assessment of treatment-induced changes at the cellular level, as these may be mediating the behavioral disturbances.

## 2. Physiological Disruptions Contributing to Chemotherapy-Related Cognitive Impairment

Multiple mechanisms contribute to cognitive impairment and hippocampal neurotoxicity in response to chemotherapy treatment, including blood–brain barrier (BBB) disruption, neuroinflammatory responses including increased pro-inflammatory cytokines (IL-6, IL-1β, TNF-α) and reduced anti-inflammatory cytokines (IL-10), reduced white matter integrity, and increases in reactive oxidative stress and mitochondrial dysfunction [14,15].

The hippocampus is particularly susceptible to insult either directly or indirectly resulting from systemic administration of various chemotherapeutic agents. Smaller hippocampal volumes following chemotherapy can be attributed to several pathophysiological changes in response to treatment. Notably, differences in neuronal morphology including reduced dendritic branching and spine density have been observed in the hippocampus, including the dentate gyrus (DG), in response to several classes of chemotherapy drugs [16,17,18,19].

Hippocampal neurogenesis is the most frequently investigated neural mechanism found to be affected by common cancer treatments, including chemotherapy [17,20,21].

Adult hippocampal neurogenesis, or the continuous addition of newborn neurons to the hippocampus during adulthood, is a phenomenon observed and extensively studied in rodents, primates, humans, and other animals [20,22,23,24,25]. There is robust evidence, based on radiocarbon cell-birth dating and extensive use of endogenous markers, that human hippocampal neurogenesis occurs throughout the entire lifespan [20,26,27,28,29,30], although, this process is more pronounced in the first years of life with a persistent decay during the course of aging. Accumulated evidence has shown that hippocampal neurogenesis plays a critical role in cognitive functions including memory consolidation [31], pattern separation [32,33], memory clearance [34,35] and cognitive flexibility [36]. All neurons added to the hippocampus during the lifetime become excitatory glutamatergic granule cells originating from a neurogenic niche located in the subgranular cell layer (SGZ) of the dentate gyrus [37]. The SGZ contains neural stem cells (NSCs) capable of self-replication, which generate neural progenitor cells (NPCs) [38,39]. NPCs also replicate and yield new NPCs in a more differentiated stage, classified as NPCs type I, II and III. After the last division, type III NPCs generate an immature granule neuron [40,41,42]. During maturation, new neurons develop dendrites and spines which receive inputs from the perforant pathway [43] and grow axons that establish synaptic contacts with CA3 [44,45]. During the last maturation stage, newborn neurons become capable of LTP formation [46,47], and after eight weeks of maturation, newborn neurons become morphologically and physiologically identical to previously formed mature granule cells [48] (Figure 1).

Chemotherapy is typically delivered systemically, with some drugs more capable of crossing the BBB, which may account for the variability in neurogenic impairment sometimes seen following treatment [49]. For drugs with low BBB permeability, the mechanisms of neurotoxicity may indirectly act through a peripheral molecular cascade. Systemic increases in pro-inflammatory cytokines contribute to breakdown of the BBB [50], allowing cytokine migration to the brain to induce an immune response, which may have a downstream impact on neurogenesis [51]. As well, chemotherapy-induced increases in reactive oxidative stress can produce cell damage that has the effect of reducing the survival of primary neural precursor cells and preventing the production of new cells [52,53,54,55], further contributing to the reduction in hippocampal neurogenesis following treatment.

Neurogenesis rates decline with age [56], in response to prolonged stress [57], and in response to anti-mitotic drugs, including many chemotherapy drugs [58]. Ki-67, a universal marker of proliferative activity [59], is especially useful for detecting neural stem cells and precursor cells undergoing division within the hippocampal SGZ [60]. Doublecortin (DCX) is a universal marker of migratory immature neurons [61] which expresses transiently in the adult DG after the last mitotic division, for approximately three weeks until the last maturation stages [62,63].

The synthetic ligand 5-bromo-2′-deoxyuridine (BrdU) allows for precise labelling of dividing cells via its incorporation into DNA [64]. Therefore, BrdU remains detectable for long periods of time following labelling. Unlike Ki-67 and DCX, BrdU provides a ‘snapshot’ of cell mitotic activity at the time of labelling [60]. Consequently, it provides an ideal method for dating the replication event that generated a specific newborn neuron. BrdU expression levels post-mortem will depend largely upon the BrdU treatment schedule (single injection or repeated daily injection) [65], and the treatment–sacrifice interval. With time, the levels of detectable cells decrease through dilution in repetitive cell divisions and apoptotic death of cells that do not reach maturation [66,67]. Short treatment–sacrifice schedules will allow for the assessment of acute proliferation rates of hippocampal neural precursor cells (similar to Ki67) [60], whereas BrdU expression levels after intervals of several weeks will be more indicative of neuronal survival rates (some co-labelling with DCX will occur in this case) [62,68].

Hippocampal toxicity has emerged as a robust finding and a major factor in CICI. In this review of the pre-clinical literature, we discuss the neurogenic and cognitive profiles associated with common chemotherapeutic agents. The focus will be on those drugs used to treat breast cancer, the type of cancer in which CICI has been most extensively studied. Specifically, we review breast cancer treatment chemotherapeutic drugs where the neurotoxic effects on hippocampal neurogenesis and related memory processing have been investigated, with attention given to identifying the methods and timing of neurogenic assessment across treatments to facilitate a comparison of protocols and findings across preclinical studies. The tables present the methodological details and highlight the heterogeneity across studies which may, in part, account for the inconsistencies reported within individual and combination chemotherapies. We highlight pre- and post-treatment interventions effective in protecting against chemotherapy-induced neurogenic depletion and memory disruption following chemotherapy treatment. Finally, we briefly relate pre-clinical findings on hippocampal disruptions observed in cancer patients following chemotherapy treatments.

## 3. Pre-Clinical Observations of Neurogenic Depletion and Memory Dysfunction Using Different Classes of Chemotherapeutic Drugs

### 3.1. Antimetabolites

The antimetabolite class of drug functions by inhibiting nucleotide synthesis. Several systemically-administered anti-metabolite drugs are permeable to the BBB, potentially gaining access to the central nervous system (CNS) via proinflammatory cytokine-induced breakdown of tight-junction proteins resulting in leaky cell junctions of the barrier [50].

#### 3.1.1. Methotrexate

Methotrexate (MTX), a folate analogue that inhibits DNA replication through inhibition of purine and pyrimidine [69], is the most frequently investigated drug associated with CICI and neurotoxicity. Cognitive impairment and neurotoxicity resulting from MTX have been shown to be relatively responsive to pharmacological interventions (see Table 1 and the interventions section below).

Seigers and colleagues [70] first identified a dosage-dependent suppression of Ki-67-labelled cells in the sub-granular zone (SGZ) three weeks following a single dose of MTX (37.5–300 mg/kg) in male rats. One month following a single high-dosage MTX treatment (250 mg/kg), rats exhibited normal performance in a standard test of spatial learning in the Morris water maze. However, the day after training, in a probe test during which the platform was removed, MTX-treated rats displayed reduced exploration of the platform’s former location as well as impairment on a test of object recognition. The latter results indicated anterograde memory loss [70]. In a follow-up study, Seigers et al. [71] assessed MTX’s effect on retrograde memory in rats premorbidly trained on the spatial water maze task or a context fear conditioning task prior to MTX administration. The spatial water maze and context fear conditioning tasks are commonly used behavioral tasks known to be sensitive to hippocampal dysfunction [72]. They found that a single MTX treatment of 250 mg/kg induced a time-dependent impairment in hippocampal cell proliferation, with reductions in Ki-67 expression emerging within seven days, but not one day, post-treatment. When tested for their pre-treatment spatial memory one month later, MTX-treated rats failed to exhibit a preference for the former platform quadrant. MTX-treated rats trained on the context fear conditioning task exhibited low rates of freezing behavior when returned to the context where they had been shocked, indicating poor remote retrograde memory following MTX treatment [71]. The neurogenic suppressing effects of a single dose of MTX appear to be limited to the first few weeks following treatment. A single high dosage of 200 mg/kg or 500 mg/kg did not produce any notable differences in hippocampal DCX or Ki-67 after just three weeks, relative to control animals. Similarly, long-term DCX levels remained comparable to controls for up to 16 weeks [73]. Notably, this time course study was conducted with mice, whereas the previous single dose investigations were conducted with rats, suggesting a potential difference in sensitivity to MTX-induced neurotoxicity across species.

Even lower dosages of MTX suppress cell proliferation and induce memory deficits. A single 40 mg/kg injection in male mice was sufficient to rapidly reduce both Ki-67 and DCX expression within 12–24 h of treatment, coinciding with a peak in TUNEL+ cells in the SGZ, indicating a surge in apoptosis. Deficits in novel object recognition memory were detected at this timepoint, indicating an early functional disruption of hippocampus-dependent memory processing despite those immature progenitor cells being incapable of functionally contributing to the hippocampal memory network at that time [74], suggesting that other cellular mechanisms are likely mediating the memory deficits during this early post-treatment time period.

In a comprehensive series of studies by Wigmore and colleagues, multiple low dosage treatments of MTX also consistently produced neurogenic and hippocampus-dependent memory disturbances. Two MTX treatments (75 mg/kg) over the course of two weeks was sufficient to induce both neurogenic suppression and memory disturbance in male rats [75,76,77,78,79]. Memory impairments in novel object and novel location memory were evident in MTX-treated mice 3–6 days following treatment [75,76,77,78]. BrdU labelling during the MTX-treatment window consistently identified impaired survival of new neurons generated during MTX treatment [75,76,77,78]. Impairments in cell proliferation and the neural progenitor population were also evident within one week of completing MTX treatment, with far fewer Ki-67+ and DCX+ cells within the SGZ compared to controls. MTX-induced suppression of cell proliferation in the SGZ at this point coincided with an increase in p21 expression [79]; p21, a CDK inhibitor, can be used as an endogenous marker of cell cycle arrest [80]. Heightened p21 expressing cells within the SGZ following MTX treatment may indicate a decreased pool of proliferating cells in the hippocampus [81,82], further depleting the neurogenic niche population through apoptosis of existing cells.

While pre-clinical investigations of chemobrain enables control over potential confounding factors such as co-morbidity, timing, and dosage of treatment, one translational limitation is that they are typically conducted in healthy animals without tumors. The presence of a tumor can independently affect peripheral physiological processes, including cognitive processing, mood and sickness behavior, inflammatory cytokine release, and even brain network activities [83,84]. To identify potential interactions between peripheral tumor development and MTX treatment on hippocampal plasticity, Seigers and colleagues [85] used an exogenous tumor cell transplantation model in advance of MTX treatment. Subcutaneous injection of hepatoma cells induced development of a localized tumor in male rats. Half of the tumor-bearing rats received a single dose of MTX 100 mg/kg^2^ weeks following tumor transplantation. While a single MTX treatment did not reduce the tumor load, MTX treatment was sufficient to reduce Ki-67-labelled cell proliferation in the hippocampus three weeks following treatment [85]. This highlights the sensitivity of the proliferative SGZ to even a single dose of MTX, while a more persistent course of treatment would likely have been required to induce an observable reduction of tumorigenic cells and overall tumor load at a clinically meaningful level.

**Table 1 ijms-22-12697-t001:** Summary of pre-clinical studies investigating the effect of methotrexate (MTX) treatment on hippocampal neurogenesis and memory processes.

Reference	Species/Strain	Sex	Age	Groups (*n*)	Dose	Treatment Schedule	Interval between Treatment and Tasks	Tasks	Chemo Behavior Memory	BrdU Schedule	NG Measures	NG Results	Intervention	Intervention Groups	Intervention Behavior	Intervention NG
Welbat et al., 2020 [79]	rat/Sprague-Dawley	M	4–5 weeks	MTX (*n* = 6) saline (propylene glycol + saline, *n* = 6)	75 mg/kg	2×; 1× on days 8 and 15	/	/	/	/	DCX	↓DCX compared to controls	hesperidin (Hsd)	100 mg/kg/day for 21 days for Hsd, Hsd + MTX (*n* = 6/each)	/	↑ DCX for Hsd groups compared to MTX alone, = DCX compared with controls
Sritawan et al., 2020 [78]	rat/Sprague-Dawley	M	4–5 weeks	MTX (*n* = 6) saline (*n* = 6)	75 mg/kg	2×; 1× on days 7 and 14	pre- (24 h) and post-treatment (5 days after)	NOR, NOL	=NOL =NOR familiarization trial; ↓NOR ↓NOL choice trial compared to controls	BrdU (1 × 100 mg/kg/day on days 5–7, 29 days before sacrifice)	BrdU, Ki67, DCX	↓BrdU, ↓Ki67, ↓DCX compared to controls	metformin	200 mg/kg/day for 14 days for metformin group, or for 14 or 28 days for the MTX + metformin groups (*n* = 6/group)	=NOL =NOR during familiarization trial; ↑NOL ↑NOR choice trial compared to MTX alone	↑BrdU, ↑Ki67, ↑DCX compared to MTX alone; =Ki67, =DCX, =BrdU compared to controls
Sirichoat et al., 2019 [76]	rat/Sprague-Dawley	M	4–5 weeks	MTX (*n* = 12) saline (*n* = 12)	75 mg/kg	2×; 1× on days 8 and 15	3 days	NOR, NOL	=NOL =NOR familiarization trial; ↓NOR and ↓NOL choice trial compared to controls	BrdU (1 × 250 mg/kg/day for 3 consecutive days starting 2 days pre-treatment)	BrdU, Ki67, DCX	↓BrdU, ↓Ki67, ↓DCX compared to controls	melatonin	8 mg/kg/day for 15 days before and during MTX, 15 days after MTX, or 30 days during and after treatment (*n* = 12/each)	=NOL =NOR familiarization trial; ↑NOR ↑NOL choice trial compared to MTX alone	=Ki67, =BrdU, =DCX compared to controls, ↑Ki67, ↑BrdU, ↑DCX compared to MTX alone
Naewla et al., 2019 [77]	rat/Sprague-Dawley	M	5 weeks	MTX (*n* = 12) saline (propylene glycol + saline, *n* = 12)	75 mg/kg	2×; 1× on days 8 and 15	/	NOR, NOL	=NOL =NOR familiarization trial; ↓NOR and ↓NOL choice trial compared to controls	BrdU (1 × 100 mg/kg/day on days 6–8)	BrdU, Ki67, DCX	↓BrdU, ↓DCX, ↓Ki67 compared to controls	hesperidin	100 mg/kg/day for 21 days for Hsd, Hsd + MTX	=NOL =NOR familiarization trial; ↑ NOL ↑ NOR choice trial compared to MTX	↑ Ki67, ↑DCX, ↑BrdU compared to MTX alone; =Ki67, =DCX, =BrdU compared to controls
Seigers et al., 2016 [73]	mouse/C57BL/6J	M	11 weeks	MTX (*n* = /) saline (*n* = /)	250 or 500 mg/kg	1×	/	/	/	/	DCX, Ki67	=DCX, =Ki67 compared to controls when sacrificed 3- and 16-weeks post-treatment	/	/	/	/
Yang et al., 2011 [74]	mouse/C57BL/6J	M	8–9 weeks	MTX (*n* = /) saline (*n* = /)	0–200 mg/kg	1×	1 and 7 days	OFT, NOR, TST (*n* = 6–8/group)	↓NOR 1- and 7-days post- treatment compared to controls (*n* = 6/group)	/	DCX, Ki67	Dose dependent:↓Ki67, ↓DCX from 0 to 12 h and maintained this low level for 14 days post-treatment (*n* = 3/group)	/	/	/	/
Lyons et al., 2011 [75]	rat/Lister Hooded	M	/	MTX (*n* = 7–12) saline (*n* = 7–12)	75 mg/kg	2×; 1× on days 1, 7	6 days	NOL	=NOL familiarization trial, ↓NOL recognition during choice trial compared to controls	BrdU (1 × 250 mg/kg on first day of chemo)	BrdU, Ki67	↓BrdU, ↓Ki67 compared to controls (*n* = 7/group)	fluoxetine (SSRI)	10 mg/kg/day for 40 days starting 1 week pre-treatment for fluoxetine and MTX + fluoxetine groups (*n* = 11–12/group)	=NOL familiarization trial, ↑NOL choice trial compared to non-fluoxetine groups, =NOL choice trial for fluoxetine groups and controls	↑BrdU, ↑Ki67 for fluoxetine alone compared to all other groups; =Ki67, =BrdU for MTX + fluoxetine and controls (*n* = 7/group)
Seigers et al., 2010 [85]	rat/Buffalo	M	9 weeks	MTX + PBS, MTX + Morris Hepatoma 7777, saline + PBS, saline + Morris Hepatoma 7777 (*n* = 7/group)	100 mg/kg	1×	/	/	/	/	KI67	↓Ki67 for saline + hepatoma 7777 compared to controls, and for MTX groups compared to controls	/	/	/	/
Seigers et al., 2009 [71]	rat/Wistar	M	12 weeks	MTX (*n* = 4–8) saline (*n* = 4–8)	250 mg/kg	1×	MWM pre- (3 days before) and post-treatment (7 days after); FC 1 month after	MWM, CFC	↓MWM ↓CFC compared to controls (*n* = 8/group)	/	Ki67	↓Ki67 when sacrificed 1 day (*n* = 4–6/group) and more 7 days (*n* = 4–8/group) post-treatment compared to controls	/	/	/	/
Seigers et al., 2008 [70]	rat/Wistar	M	12 weeks	MTX Saline	[e1] 37.5, 75, 150 or 300 mg/kg, [e2] 250 mg/kg	1x	[e2] 3–4 weeks	[e2] MWM, NOR	[e2] MWM = escape latency and ↓latency to cross; ↓NOR compared to controls (*n* = 8/group)	/	[e1] Ki67	[e1] Dose dependent:↓Ki67 compared to controls (*n* = 6/group/dose)	/	/	/	/

Abbreviations: 5-fluorouracil (5FU); bilateral non-stimulation (BNS); 5-bromo-2′-deoxyuridine (BrdU); bilateral stimulation (BS); conditional associative learning (CAL); contextual conditioned response (CER); context fear conditioning (CFC); cued memory (CM); cyclophosphamide (CPP); doublecortin (DCX); dentate gyrus (DG); discrimination learning (DL); doxorubicin (DOX); docetaxel (DTX); experiment (e); environmental enrichment (EE); elevated plus maze (EPM); female (F); forced swim task (FST); hesperidin (Hsd); kilogram (kg); lithium (Li); male (M); milligram (mg); methotrexate (MTX); Morris water maze (MWM); novel location recognition (NLR); nonmatching-to-sample test (NMTS); novel object location (NOL); novel object recognition (NOR); non-significant (NS); novelty-suppressed feeding (NSF); open field (OF); passive avoidance test (PA); postnatal day (PD); probe test (PT); paclitaxel (PTX); standard environment (SE); subgranular zone (SGZ); spatial memory (SM); sucrose preference (SP); temozolomide (TMZ); tail suspension task (TST) very long delay conditioning (VLD); water for injection (WFI); Y-maze (YM); no difference (=); increase (↑); decrease (↓).

#### 3.1.2. 5-Fluorouracil

5-Fluorouracil’s (5-FU) primary mechanism of action is the inhibition of thymidine synthesis and blocking DNA replication [86]; 5-FU is capable of diffusing across the BBB [87,88] and directly impacting mitotic activity in the brain. It is among the most common chemotherapeutic drugs found to have long-lasting neurogenic toxicity. Early studies found that three systemic treatments of 5-FU (40 mg/kg) over the course of five days impaired long-term survival of adult-generated neurons for up to six months after 5-FU treatment in young adult mice, whereas BrdU labelled cells were comparable to controls 1 and 7 days following treatment [89]. This suggests that suppressed hippocampal cell proliferation rates may not be immediately evident following 5-FU treatment, while long term survival of post-treatment generated neurons is impaired.

In adult male rats, five systemic treatments of 20 mg/mg 5-FU over the course of 12 days slightly impaired subsequent memory in a novel object location recognition task, with 5-FU treated rats expressing no preference for the novel location following a 5 min interval [90]. Ki-67-expression assessed immediately following behavioral testing revealed no difference in the number of proliferating cells in the hippocampus relative to control-treated rats, in line with observations of Han et al. [89], who found that cell proliferation in the SGZ is not immediately impacted by systemic 5-FU treatment. They did find that 5-FU treatment led to a reduction in both DCX and brain-derived neurotrophic factor (BDNF) protein within the hippocampus, which they proposed may lead to a reduction in neural differentiation of adult-generated precursor cells given the putative role of BDNF in promoting cell differentiation [91]. Similarly, three daily treatments of 50 mg/kg 5-FU did not identify differences in cell proliferation in young adult mice when assessed 2 h following the final 5-FU + BrdU treatment. This null effect of 5-FU treatment on hippocampal cell proliferation fits with the temporal profile of a lack of early treatment impairment. However, a small sample size potentially limited statistical power to detect differences between treatment conditions [92] (Table 2).

When young adult mice were given a higher dose treatment regimen of four 60 mg/kg 5-FU injections within seven days followed by BrdU labelling, a modest reduction of 15% of proliferative cells was observed after just 24 h [93]. This suggests that higher toxicity levels of 5-FU have a more immediate impact on cell proliferation rates, while lower doses are less immediately neurotoxic to cell replication. In line with evidence for a cumulative cytotoxic effect of 5-FU, a single high dosage treatment of 75 mg/kg 5-FU did not result in differences in Ki-67 or DCX expression in DG when assessed 3 and 16 weeks post-treatment [73].

In male rats, BrdU labelling of dividing cells prior to five systemic treatments of 25 mg/kg 5-FU over two weeks was sufficient to strongly reduce the one month survival of new neurons in the SGZ, to impair novel object location recognition memory, and to suppress context-dependent emotional suppression memory one month following 5-FU treatment. In addition to impaired survival of pre-treatment generated neurons, post-treatment proliferation of Ki-67-expressing cells was also reduced in response to 5-FU [94,95], consistent with the longer time course of 5-FU-mediated neurogenic suppression in response to lower doses but cumulative treatments.

Given the higher incidence of common cancers such as breast and prostate cancer in older adults, it is important to understand the neurotoxic profile of common chemotherapeutic agents across the lifespan. Dubois et al. [96] assessed the long-term cytotoxic effects of 5-FU for two months post-treatment in young and aged male mice. Three weekly injections of 37.5 mg/kg 5-FU did not adversely affect spatial learning in a water maze in young or older mice. However, 5-FU did impair reversal learning of a new platform location in both age groups, indicating a deficit in behavioral flexibility or perseveration in an established spatial strategy. BrdU labelling of cells prior to sacrifice identified a large decrease in BrdU expression within the SGZ in younger mice, but no detectable reduction in aged mice. This lack of a treatment effect in aged mice is confounded by an overall age-related reduction in basal neurogenic rates, making any further reductions impossible to detect [96].

Most recently, a treatment protocol of five injections of 25 mg/kg 5-FU in male rats over two weeks strongly suppressed one month survival of BrdU+ cells labelled prior to 5-FU treatment. Persistent reductions in both Ki-67 and DCX-expressing cells were also observed several weeks following the end of 5-FU treatment, indicative of long-term suppression of cell proliferation [97]. At this timepoint, 5-FU treated rats displayed impaired memory for the novel object location task, indicative of spatial memory impairment [97]. Using this same treatment protocol, this group also identified a reduction in p21 expressing cells in the SGZ four days [98] and one month [99] following the end of 5-FU treatment, indicating both early and persistent disruption of mitotic activity and apoptosis. p21 may be an early indication of arrested mitotic activity that is not captured at the early post-treatment timepoint by conventional endogenous protein markers of neurogenesis Ki-67 and DCX. 

**Table 2 ijms-22-12697-t002:** Summary of pre-clinical studies investigating the effect of 5-Fluorouracil (5-FU) treatment on hippocampal neurogenesis and memory processes.

Reference	Species/Strain	Sex	Age	Groups (*n*)	Dose	Treatment Schedule	Interval between Treatment and Tasks	Tasks	Chemo Behavior Memory	BrdU Schedule	NG Measures	NG Results	Intervention	Intervention Groups	Intervention Behavior	Intervention NG
Suwannakot et al., 2021 [98]	rat/Sprague-Dawley	M	adult	5-FU (*n* = 7) saline (*n* = 7)	25 mg/kg	5×; 1× on days 9, 12, 15, 18, and 21	/	/	/	/	DCX	↓DCX compared to controls	melatonin	8 mg/kg/day for 21 days (melatonin, melatonin + 5-FU)	/	=DCX compared to controls, ↑ DCX compared to 5-FU alone
Sirichoat et al., 2020 [97]	rat/Sprague-Dawley	M	4–5 weeks	5-FU (*n* = 12) saline (*n* = 12)	25 mg/kg	5×; 1× on days 9, 12, 15, 18, and 21	25 days	NOL	=NOL familiarization trial, NS ↓NOL choice trial compared to controls	BrdU (1×/day on days 7–9)	BrdU, Ki67, DCX	↓BrdU, ↓Ki67, ↓DCX compared to controls	melatonin	8 mg/kg/day for 21 days (melatonin, melatonin + 5F-FU), or 42 days (melatonin + 5-FU) (*n* = 12/group)	=NOL familiarization and choice trials compared to controls, ↑NOL for melatonin groups compared to 5-FU alone	↑BrdU ↑Ki67 ↑DCX compared to 5-FU alone, ↓BrdU, ↓Ki67, ↓DCX for melatonin + MTX groups compared to controls and melatonin alone
Welbat et al., 2018 [99]	rat/Sprague-Dawley	M	4–5 weeks	5-FU (*n* = 10) saline (propylene glycol + saline, *n* = 10)	25 mg/kg	5×; 1× on days 8, 11, 14, 17, and 20	/	/	/	/	DCX	↓DCX compared to controls	Asiatic acid (AA)	30 mg/kg/day on days 1–20 or 21–40 (AA + 5-FU), or 1–40 (AA) (*n* = 10/group)	/	↑DCX for all AA groups except compared to 5-FU alone, except =DCX for recovery and 5-FU alone
Seigers et al., 2016 [73]	mouse/C57BL/6J	M	11 weeks	5-FU (*n* = /) saline (*n* = /)	75 mg/kg	1x	/	/	/	/	DCX, Ki67	=DCX, =Ki67 compared to controls when sacrificed 3- and 16-weeks post-treatment	/	/	/	/
Dubois et al., 2014 [96]	[e1 and e3] mouse/C57BL/6J	[e1 and e3] M	[e1] 8 weeks (juvenille) and 20 months (adult)	[e1] 5-FU (*n* = 10–12/group) saline (*n* = 12–13/group)	[e1 and e3] 37.5 mg/kg	[e1 and e3] 3×; 1× on days 0, 7, 14	[e1 and e3] 24 days	[e1 and e3] EPM, FST, MWM, NOR	[e1] =MWM ↑NOR compared to controls, ↓MWM ↓NOR for adults compared to juveniles	BrdU (50 mg/kg 47 post-treatment × 2/13 h intervals [e1] or 1×/day for 4 consecutive days [e3]	[e1 and e3] BrdU	[e1] ↓BrdU for young 5-FU compared to young controls, =BrdU among aged groups, ↓BrdU for adults compared to juveniles	[e3] glucose or WFI	[e3] 5% glucose or WFI 3×, 7 h before each treatment (WFI or glucose/Saline or 5-FU in young mice, *n* = 11–14/ group)	[e3] ↑NOR for 5-FU/WFI compared to saline/WFI and glucose groups, =MWM for glucose groups and saline/WFI	[e3] ↓BrdU for saline/WFI compared to saline/glucose, =BrdU between 5-FU/WFI and glucose, NS ↓BrdU for 5-FU/glucose or WFI compared to saline/glucose or WFI
ElBeltagy et al., 2010 [95]	rat/Lister Hooded	M	/	5-FU (*n* = 12) saline (*n* = 12)	20 mg/kg	6×; 1× every 2 days for 2 weeks	/	NOL	=NOL familiarization trial, ↓NOL choice trial	/	Ki67	↓Ki67 compared to controls (*n* = 7–8/group)	fluoxetine (SSRI)	10 mg/kg/day over three weeks (fluoxetine, 5-FU + fluoxetine, *n* = 10–11/ group)	=NOL familiarization trial, ↑NOL choice trial for fluoxetine and controls compared to 5-FU + fluoxetine	=Ki67 between fluoxetine groups and controls, ↑Ki67 compared to 5-FU alone (*n* = 7/group)
Lyons et al., 2012 [94]	rat/Lister Hooded	M	/	5-FU (*n* = 12) saline (*n* = 12)	25 mg/kg	5×; 1× on days 8, 11, 14, 17, and 20	7 days after the last fluoxetine treatment	NOL	=NOL familiarization trial, ↓NOL choice trial compared to controls	BrdU (100 mg/kg/day on days 6–8)	BrdU, Ki67	↓BrdU, ↓Ki67 compared to controls	fluoxetine (SSRI)	10 mg/kg/day for 20 days starting 5 days before first BrdU, 40 days before and during treatment, or 20 days starting the last day of treatment (fluoxetine + 5-FU), or 40 days (fluoxetine) (*n* = 12/group)	=NOL familiarization trial, ↑NOL choice trial for fluoxetine groups except =NOL choice trial for the recovery group and 5-FU alone	↑Ki67, ↑BrdU compared to 5-FU alone except =BrdU for 5-FU alone and recovery group; ↑Ki67, ↑BrdU for fluoxetine alone compared to controls; =Ki67 =BrdU for 5-FU + fluoxetine groups compared to controls except for recovery group
Janelsins et al., 2010 [93]	mouse/C57BL/6J	/	6–8 weeks	5-FU (*n* = 6) saline (*n* = 8)	60 mg/kg	3×; 1× on days 1, 4, 7	/	/	/	BrdU (4 × 50 mg/kg/2 h intervals, 24 h post-treatment)	BrdU	↓BrdU compared to controls	/	/	/	/
Mustafa et al., 2008 [90]	rat/Lister Hooded	M	adult	5-FU (*n* = 9) saline (*n* = 8)	20 mg/kg	5×; over 12 days	/	NOL	=NOL familiarization trial, ↓NOL choice trial compared to controls	/	Ki67, DCX	=KI67, ↓DCX compared to controls	/	/	/	/
Han et al., 2008 [89]	mouse/CBA	/	6–8 weeks	5-FU (*n* = 5) saline (*n* = 5)	40 mg/kg	3×; 1× on days 1, 3, 5	/	/	/	BrdU (1 × 50 mg/kg, 4 h before perfusion)	BrdU, DCX	↓BrdU on day 14 to 6 months; ↓DCX when sacrificed 1 and 56 days post-treatment compared to controls	/	/	/	/
Mignone & Weber., 2006 [92]	mouse/C57BL/6J	/	6 weeks	5-FU (*n* = 3) saline (*n* = 3)	50 mg/kg	3×; 1× on days 1, 2, 3	/	/	/	BrdU (1 × 200 mg/kg with third chemo injection)	BrdU	=BrdU compared to controls	/	/	/	/

#### 3.1.3. Cisplatin

Cisplatin facilitates DNA cross linking, inhibiting DNA replication and causing cell death in dividing cells. Cisplatin easily passes through the BBB following systemic administration [100]. Dietrich and colleagues [101] were amongst the first to verify neurogenic depletion as a candidate neural mechanism underlying cisplatin-induced hippocampus-dependent memory dysfunction in rodent models (Table 3). Young adult mice received three injections of 5 mg/kg cisplatin over five days, and proliferating cells were assessed 1 and 42 days later. BrdU labelling of actively dividing cells 4 h before sacrifice identified an immediate suppression of proliferating cells in the SVZ and a return to control levels by day 42, suggesting a transient impairment of new cell generation in response to cisplatin [101]. TUNEL staining to identify apoptotic cells within the hippocampus identified an immediate increase in cell death lasting at least 42 days post-treatment, with the majority of TUNEL+ cells co-localizing with DCX. This indicates that, despite seemingly normal proliferation rates at this timepoint, recently proliferated cells are likely non-viable following cisplatin treatment, contributing to a long-term impairment of neuroblast cells [101]. A single dose of 12 mg/kg cisplatin in rats similarly identified a rapid suppression of Ki-67-labelled proliferating cells in the SGZ just two days following treatment at this higher dosage. At this time, gene expression of several pro-apoptotic genes within the Bcl2 family was found in the hippocampus, but not in a control assay within the anti-proliferative superior colliculus [102]. Their findings complement those of Dietrich et al. [101], relating the chemosensitivity of neural precursor cells and cell death within the SGZ to a decline in the neurogenic niche available for neuronal differentiation shortly following systemic cisplatin treatment.

Hinduja at al. [103] confirmed that a single systemic cisplatin dose of 12 mg/kg continued to suppress DCX-expression for one week in rats. Similar findings have been reported by several groups administering repeated cycles of five weekly injections at smaller dosages. Male mice received two or three cycles of low-dosage cisplatin injections (2.3 mg/kg) over one month. Cognitive assessment one week later identified deficits in novel object location memory, reduced exploratory behavior in a Y-maze [104], and longer escape latencies in the water maze and object memory impairments [105], indicative of hippocampus-dependent memory deficits. DCX expression within the SGZ was drastically suppressed in response to cisplatin treatment immediately [105] and one week following the end of cisplatin treatment [104]. A substantial depletion of dendritic spine density was also observed following three cycles of low dosage cisplatin treatment [105], indicating that cisplatin additionally damages the structural and functional integrity of mature hippocampal neurons.

**Table 3 ijms-22-12697-t003:** Summary of pre-clinical studies investigating the effect of cisplatin treatment on hippocampal neurogenesis and memory processes.

Reference	Species/Strain	Sex	Age	Groups (*n*)	Dose	Treatment Schedule	Interval between Treatment and Tasks	Tasks	Chemo Behavior Memory	BrdU Schedule	NG Measures	NG Results	Intervention	Intervention Groups	Intervention Behavior	Intervention NG
Yi et al., 2020 [105]	mouse/SPF C57BL/6J	M	8 weeks	cisplatin (*n* = 3–15) saline (*n* = 3–15)	34.5 mg/kg	5×; 1× on days 1–5, 11–15, and 21–25	/	OF, NOR, MWM	↓MWM ↓NOR compared to controls (*n* = 15/group)	/	DCX	↓DCX compared to controls (*n* = 3/group)	curcumin	100 mg/kg 1 h pre-treatment to control + curcumin, cisplatin + curcumin groups	↑MWM ↑NOR compared to cisplatin without curcumin, =MWM for saline + curcumin and controls (*n* = 15/each)	↑DCX compared to cisplatin without curcumin group (*n* = 3/group)
Chiu et al., 2017 [104]	mouse/C57BL/6J	M	/	cisplatin (*n* = 8–12) saline (*n* = 8–12)	2.3 mg/kg	10×; 1× on days 1–5, and 11–15	7 days	NOR, YM, FST, SP	↓YM (*n* = 4/group); ↓NOR compared to controls (*n* = 8/group)	/	DCX	↓DCX compared to controls (*n* = 12/group)	Pifithrin-u (PFT-u)	8 mg/kg administered 1 h before cisplatin (PFT-u, PFT-u + cisplatin)	↑YM (*n* = 4/group) and ↑NOR (*n* = 8/group) compared to without PFT-u	↑DCX compared to without PFT-u (*n* = 12/group)
Hinduja et al., 2015 [103]	rat/Sprague Dawley	M	12 weeks	cisplatin (*n* = 3) saline (*n* = 3)	12 mg/kg	1×	/	/	/	/	DCX	↓DCX 2 days and more so 7 days post-treatment, =DCX after 21 days compared to controls	D-methionine	30 mg/mL administered 30 min prior to chemo (D-methionine, D-methionine + cisplatin) (*n* = 3/group)	/	↑DCX for D-methionine groups compared to cisplatin alone, =DCX for D-methionine and controls
Manohar et al., 2014 [102]	rat/Sprague Dawley	M	/	cisplatin (*n* = 3) saline (*n* = 3)	12 mg/kg	1×	/	/	/	/	Ki67	↓Ki67 2 days post-treatment compared to controls	/	/	/	/
Dietrich et al., 2006 [101]	mouse/CBA	/	6–8 weeks	cisplatin (*n* = 5) saline (*n* = 5)	5 mg/kg	3×; 1× on days 1, 3, 5	/	/	/	BrdU (1 × 50 mg/kg administered 4 h before perfusion)	BrdU	↓BrdU 1- day and NS ↓BrdU 42-days post-treatment compared to controls	/	/	/	/

#### 3.1.4. Cytarabine

Cytarabine (Ara-C), a cystine analogue that incorporates into DNA and prevents subsequent DNA and RNA replication [106], is most often used to treat various forms of leukemia but is also used as a breast cancer treatment (Table 4). The neurogenic effects of systemically administered cytarabine have not been extensively studied. However, in one investigation, young adult mice were given three injections of 250 mg/kg cytarabine over five days, and proliferating cells were assessed immediately or up to 56 days later to identify cytarabine’s short and long-term cytotoxicity profile within the hippocampus. BrdU labelling of actively dividing cells 4 h before sacrifice at each timepoint identified a time-dependent decline in hippocampal cell proliferation, with the most significant reduction seen two months following treatment. No effects were detected after one week of treatment. Conversely, TUNEL staining to identify apoptotic cells in the hippocampus revealed the reverse temporal pattern, with high levels of TUNEL+ cells up to two weeks following treatment, with levels comparable to controls 56 days later [101]. This pattern of cell proliferation and cell death suggests that hippocampal cells may remain capable of actively dividing following cytarabine treatment, however, a simultaneous boost in rates of apoptosis contributes to hippocampal depletion and instability of the neural architecture at an early post-treatment period. Over time, as rates of cell death return to normal, an emerging deficit in the renewal of the neuronal population could result in a less plastic hippocampal neural network that would impact learning and memory [107]. Of the existing mature hippocampal neuronal population, cytarabine has been linked to lower dendritic arborization and spine density within the DG and downstream CA3 and CA1 sub-regions, indicating that the existing neuronal pool may be morphologically underdeveloped, contributing to spatial memory deficits observed in response to cytarabine treatment [108].

### 3.2. Alkylating Agents

Alkylating agents, among the oldest classes of chemotherapeutic drugs used in cancer treatments [109], act directly on DNA to induce cross linking of DNA strands, preventing transcription, DNA stand breaks, abnormal base pairing, and eventually leading to cell death [110].

#### 3.2.1. Cyclophosphamide

Cyclophosphamide (CPP) is amongst the most commonly studied anti-cancer drugs found to induce neurogenic depletion in both mice and rats [17,111] (Table 5). The neurogenic suppressing effects of CPP were first identified by Janelsins et al. [93], who sought to compare the neurotoxic effects of common chemotherapies known to cross the BBB, including CPP, with those that are restricted from crossing into the CNS. Young adult mice were given three 50 mg/kg treatments of CPP over the course of one week, and proliferating cells were labelled with BrdU one day following the final CPP injection. Brains collected 24 h following BrdU treatment revealed a rapid 30% reduction in cell proliferation in the SGZ [93]. Similarly, four weekly CPP treatments at a higher dosage of 80 mg/kg [112] resulted in an approximately 40% reduction of DCX-labelled cells and a 30% loss of BrdU-labelled cells just 2 h following the final CPP treatment, confirming rapid neurogenic suppression following treatment, with multiple treatments.

Four weekly treatments of 50 mg/kg CPP were found to inhibit neurogenic rates and impair hippocampus-dependent novel object location memory [17,113], spatial learning and memory in the water maze [114], and 24 h context fear memory in male rats [17]. BrdU labelling following the final CPP treatment reduced the one- and four-week survival of adult-generated hippocampal cells, and suppressed immature DCX-labelled neurons in the DG four weeks following treatment. Notably, new neurons generated following CPP treatment developed abnormal dendritic morphology including shorter dendritic length, less branching, thinner dendritic shafts, lower spine density, and ectopic migration away from the granule cell layer and into the hilus [17,114]. These observations suggest that new cells produced following CPP treatment will not normally incorporate into the DG neuronal network, causing functional disruptions of neuronal signaling and downstream signaling to the CA3.

A more intensive but lower dose treatment schedule of seven doses of 30 mg/kg CPP over two weeks did identify a reduction in survival of neurons BrdU-labelled on the first day of CPP treatment, confirming that cells dividing during the CPP treatment do not develop into viable neurons [111]. These authors failed to find a difference in Ki-67-expressing cells one week following the final CPP treatment, indicating an eventual return to normal post-treatment cell proliferation rates using a lower CPP dosage in rats [111].

A single CPP treatment does not cause the same level of persistent impairment as observed after multiple doses, suggesting the neurotoxic effect of CPP is cumulative and potentially dose-dependent. Cell proliferation deficits were confirmed in male mice following a single dose of 40 mg/kg [115], where both DCX and Ki-67 expression in DG were suppressed 24 h following single CPP treatment but returned to baseline levels ten days following treatment. The time window for identifying CPP-induced neurogenic depletion is brief, as a single CPP treatment at a high dose of 150 mg/kg did not result in observed differences in Ki-67 or DCX expression levels in DG when assessed after 3 and 16 weeks following the high dosage treatment [73]. The transient reduction in cell proliferation indicates that a single CPP treatment likely does not have robust deleterious effects on hippocampal plasticity in mice, and that CPP may only acutely impair hippocampal cell proliferation in mice, with additive effects over repeated treatments.

**Table 5 ijms-22-12697-t005:** Summary of pre-clinical studies investigating the effect of cyclophosphamide (CPP) treatment on hippocampal neurogenesis and memory processes.

Reference	Species/Strain	Sex	Age	Groups (*n*)	Dose	Treatment Schedule	Interval between Treatment and Tasks	Tasks	Chemo Behavior Memory	BrdU Schedule	NG Measures	NG Results	Intervention	Intervention Groups	Intervention Behavior	Intervention NG
Wu et al., 2017 [114]	rat/Sprague Dawley	M	6–8 weeks	CPP (*n* = 3–7) saline (*n* = 3–7)	25 or 50 mg/kg	4×; 1/week for 4 weeks	4 weeks	MWM	↓MWM compared to controls (*n* = 7/group)	/	DCX	Dose dependent: ↓DCX compared to controls (*n* = 3/group)	/	/	/	/
Seigers et al., 2016 [73]	mouse/ C57BL/6J	M	11 weeks	CPP (*n* = /) saline (*n* = /)	150 mg/kg	1×	/	/	/	/	DCX, Ki67	=DCX, =Ki67 compared to controls when sacrificed 3- and 16-weeks post-treatment	/	/	/	/
Kitamura et al., 2015 [113]	rat/Wistar	M	/	CPP (*n* = 6) saline (*n* = 6)	50 mg/kg	4×; 1/week for 4 weeks	SP 1 day before and 7 days post-treatment; other tasks 7 days post-treatment	light-dark test, NOL, SP	↓ NOL recognition compared to controls	BrdU (4 × 50 mg/kg/6 h intervals) either 7 days post-treatment (cell proliferation) or 24 h before last treatment (cell survival)	BrdU	↓BrdU for cell proliferation and survival compared to controls	/	/	/	/
Hou et al., 2013 [112]	mouse/ICR	M	8 weeks	CPP (*n* = 8) saline (*n* = 8)	80 mg/kg	4×; 1/week for 4 weeks	1 day	YM, PA	↓YM, ↓PA compared to controls	BrdU (1 × 100 mg/kg/day for 3 days, starting 1-day post-treatment)	BrdU, DCX	↓DCX, ↓BrdU compared to controls	Ginsenoside Compound K	CPP + 2.5, 5, or 10 mg/kg of Ginsenoside Compound K (*n* = 8/each)	=YM =PA for Ginsenoside Compound K groups and controls	Dose-dependent ↑ DCX, ↑ BrdU compared to CTX alone
Christie et al., 2012 [17]	rat/Athymic Nude	M	8 weeks	CPP (*n* = 10) saline (*n* = 8)	50 mg/kg	4×; 1/week for 4 weeks	7 days	NOL, CFC	↓NOL, ↓CFC in the context test, except =freezing response in all other tests compared to controls	BrdU (1 × 100 mg/kg/day for 6 days, starting 2 days post-treatment)	BrdU, BrdU-NeuN, DCX	NS ↓BrdU ↓BrdU-NeuN, ↓DCX compared to controls	/	/	/	/
Lyons et al., 2011 [111]	rat/Lister-hooded	M	/	CPP (*n* = 12) saline (*n* = 12)	30 mg/kg	7×; every 2 days for 2 weeks	5 days	NOL	=NOL recognition during familiarization and choice trials compared to controls	BrdU (250 mg/kg after the first treatment)	BrdU, Ki67	↓BrdU, =Ki67 compared to controls	/	/	/	/
Yang et al., 2010 [115]	mouse/ICR	M	8–10 weeks	CPP (*n* = 6–9) saline (*n* = 6–9)	40 mg/kg	1×	12 h and 10 days	PA, foot shock, NOR	↓PA ↓NOR 12 h and =PA =NOR 10 days post-treatment compared to controls (*n* = 9/group)	/	DCX, Ki67	↓DCX, ↓Ki67 24 h and =DCX, =Ki67 2–10 days post-treatment compared to controls (*n* = 6/group)	/	/	/	/
Janelsins et al., 2010 [93]	mouse/C57BL/6J	/	6–8 weeks	CPP (*n* = 6) saline (*n* = 8)	50 mg/kg	3×; 1× on days 1, 4, 7	/	/	/	BrdU (4 × 50 mg/kg/2 h intervals, 24 h post-treatment)	BrdU	↓BrdU compared to controls	/	/	/	/

#### 3.2.2. Carmustin

Carmustin (BCNU) is an alkylating agent most commonly used in treating glioma, but has also been used in combination therapy for breast cancer involving brain metastases [116,117]. In one study comparing the neurotoxic effects of carmustin against several other common chemotherapeutic agents, young adult mice, given three injections of 10 mg/kg carmustin over five days developed a four-fold increase in TUNEL+ cells in the DG for up to ten days following the final carmustin treatment. Co-labelling of these cells confirmed that apoptosis occurred primarily in DCX+ neural progenitor cells. A dose-dependent decline in survival of neural progenitor cells was also confirmed in vitro in response to exposure to sub-lethal doses of carmustin. BrdU labelling 2 h prior to sacrifice identified control levels of cell proliferation one day after carmustin treatment, suggesting normal neurogenic activity soon after carmustin treatment [101]. However, the parallel finding of apoptotic DCX+ cells at that time indicates that the recently proliferated BrdU+ cells will not likely survive. Rates of apoptosis within the DG declined back to control levels after six weeks, however, this coincided with a reduction of BrdU+ cells at this timepoint [101]. Together, these findings identify that the early apoptosis in neural progenitors and the long-term decrease in cell proliferation following carmustin treatment have cumulative negative effects on hippocampal volume and cognitive function (Table 6). 

#### 3.2.3. Temozolomide

Temozolomide (TMZ), another DNA alkylating agent with high BBB permeability [118], has been used as a method of intentionally suppressing adult neurogenesis in rodents to investigate the role of post-natally generated hippocampal neurons in memory processing through targeted loss-of-function approaches. Perhaps because of their non-clinical focus, these studies, discussed below, typically have not been included in pre-clinical reviews of CICI [15,21,51] (Table 7).

TMZ was first found to suppress BrdU-labelled cells in the DG in a dose-dependent manner, with reductions up to 80% in cell proliferation seen following four weekly treatments of 10, 25, and 50 mg/kg for three consecutive days [119]. TMZ-induced neurogenic ablation was associated with reduced LTP within the DG, but did not impact downstream LTP in CA1. Hippocampus-dependent spatial memory in the water maze task was impaired in TMZ-treated mice, as was new spatial learning following reversal platform training [119,120]. Stone and colleagues [121] used the same TMZ treatment protocol and similarly found that four weeks of 25 mg/kg of TMZ was sufficient to reduce six-week survival of new neurons in young mice, but did not observe spatial memory deficits when using a weak training protocol, which is not typically sufficient to support robust spatial memory formation even in normal mice [122]. Four weeks of TMZ treatment were also sufficient to reduce Ki-67-expression in the SGZ and to impair context discrimination memory 24 h after acquiring a fear memory in a distinct context in mice [123].

TMZ-induced neurogenic attenuation is observed across the lifespan, with rates of suppression being proportional to the rate of proliferation. Juvenile mice typically have high basal rates of post-natal hippocampal cell proliferation relative to adult mice, with proliferation rates declining rapidly with age [124,125,126]. BrdU injection following the completion of four weeks of TMZ treatment revealed a 70–80% reduction of BrdU labelled cells in the SGZ relative to non-treated juvenile mice (1–2 months old), young adult (3 months old) and middle-aged mice (12 months old) [35]. Notably, juvenile mice have significantly higher basal rates of neurogenesis relative to middle aged mice. As a result, an 80% reduction following TMZ treatment in young animals has a proportionally greater impact on the functional hippocampal circuitry. Accordingly, when mice were subsequently trained on the spatial water maze task using a strong training protocol, which typically supports robust spatial memory in control mice [122], spatial memory deficits were observed in juvenile mice, but not in adult mice. These findings predict that chemotherapeutic treatments in younger patients will have more severe consequences on their neural development and hippocampal integrity, as has been observed in juvenile cancer survivors [20,127,128]. While TMZ-mediated suppression of hippocampal neurogenesis is detrimental to the establishment of new hippocampus-dependent memories in young mice, paradoxically, TMZ-induced suppression of hippocampal cell proliferation prevents forgetting of previously established context memories, possibly by preventing the dynamic remodeling of existing DG neural networks and synaptic connections supporting existing memory traces, leading to the persistence of previously acquired memories in the young brain [129,130,131].

TMZ has been shown to induce neurogenic depletion using a variety of treatment protocols. Five weeks of 25 mg/kg TMZ treatment in adult rats reduced survival of BrdU-labelled hippocampal neurons generated during the first week of treatment and reduced survival of neurons labelled after three treatment cycles [132]. Trace eye blink conditioning, a hippocampus-dependent task [133], was unaffected after only one cycle of treatment. At this time, behavioral or cognitive impairment resulting from suppressed newborn neurons would not be expected, as the remaining population of unaffected hippocampal neuronal network could compensate in processing the memory task. Deficits in trace eyeblink conditioning were detected after multiple rounds of TMZ treatment, at a time when prolonged neurogenic suppression compromised the network’s ability to form new memory traces in the absence of input from functional newly generated neurons. In line with this altered population dynamic interpretation, four weeks of TMZ treatment attenuated spontaneous theta activity in the dorsal DG, a finding not seen after only one week of TMZ treatment [132].

TMZ-induced neurogenic depletion is not equally observed throughout the hippocampus. Regional differences in neurogenic activity were found across the hippocampal long axis following six weeks of 25 mg/kg TMZ treatment. Systemic TMZ treatment persistently suppressed DCX and Ki-67 expression throughout both the dorsal and ventral DG in young adult mice, persisting nine weeks following the end of treatment. Cell proliferation in the ventral portion of the hippocampus was more severely impacted, with a reduction of 35% relative to control levels, while proliferation in the dorsal hippocampus was reduced by 31% [134]. A brief TMZ treatment protocol of only three daily injections (25 mg/kg) was sufficient to induce a slight impairment in 24 h context fear memory along with impairments in novel object location memory and social recognition memory when assessed four days following TMZ treatment. Similar to Egeland et al. [134], daily BrdU injections throughout the three-day treatment window confirmed a reduction in proliferation of cells in both the dorsal and ventral DG [135]. 

**Table 7 ijms-22-12697-t007:** Summary of pre-clinical studies investigating the effect of temozolomide (TMZ) treatment on hippocampal neurogenesis and memory processes.

Reference	Species/Strain	Sex	Age	Groups (*n*)	Dose	Treatment Schedule	Interval between Treatment and Tasks	Tasks	Chemo Behavior Memory	BrdU Schedule	NG Measures	NG Results	Intervention	Intervention Groups	Intervention Behavior	Intervention NG
Pereira-Caixeta et al., 2018 [135]	mouse/Swiss	M	8–12 weeks	TMZ (*n* = /) saline (*n* = /)	25 mg/kg	3×; 1× on days 1–3	5 days	EPM, OF, NOL, CFC, MWM, social recognition test	↓ NOL ↓freezing in CFC ↓social recognition compared to controls (*n*= 10/group)	BrdU (1 × 75 mg/kg/day for 7 days)	BrdU, BrdU-NeuN	↓BrdU, ↓BrdU-NeuN in the ventral and dorsal HPC (*n* = /)	/	/	/	/
Egeland et al., 2017 [134]	mouse/C57BL/6J	M	10 weeks	TMZ (*n* = 10) saline (*n* = 10)	25 mg/kg	18×; 1× on 3 consecutive days every week for 6 weeks	6 weeks	OF, EPM, TST, FST, SP, NSF	/	/	Ki67, DCX	↓DCX in ventral but NS ↓DCX in dorsal DG, ↓Ki67 in the ventral and dorsal DG, ↓DG volume compared to controls	/	/	/	/
Akers et al., 2014 [129]	mouse/C57BL/6NTac x 129SvEvTac	M, F	PD17	TMZ (*n* = 7–22) saline (*n* = 7–22)	25 mg/kg	16×; 1× on 4 consecutive days every week for 4 weeks	TMZ after training but before shock and test sessions; or TMZ before training, shock, and test sessions.	CFC	=CFC freezing for TMZ before testing; ↑CFC for shocked TMZ compared to controls and non-shocked TMZ (*n* = 22/ group)	/	Ki67, DCX	↓Ki67 (*n* = 10–12/group),↓DCX compared to controls (*n* = 7/group)	/	/	/	/
Martinez-Canabal et al., 2013 [35]	mouse/C57BL/6NTac x 129SvEvTac	/	1, 2, or 11 months	TMZ (*n* = 6–17/group) saline (DMSO, *n* = 6–16/group)	25 mg/kg	12×; 1× on 3 consecutive days every week for 4 weeks	1 day	MWM	↓MWM compared to controls, for middle-aged compared to other age groups, for juveniles compared to adults; ≠ group differences among adults or middle-aged mice	BrdU (1 × 200 mg/kg, 24 h post-chemo)	BrdU	↓BrdU in all three age groups compared to controls	/	/	/	/
Nokia et al., 2012 [132]	rat/Sprague Dawley	M	8–10 weeks	TMZ (*n* = /) saline (*n* = /)	25 mg/kg	1× on 3 consecutive days every week [e1] 12× for 4 weeks; [e2 and e4] 15× for 5 weeks; [e3] 18× for 6 weeks	1 day	delay and trace eye blink conditioning, trace and VLD	Compared to controls: [e2] ↓trace and ↓delay conditioning on the first day but = delay afterwards [e3] =VLD ↓trace [e3 and e4] = trace conditioning short-term but ↓trace long-term (*n* = 5–9/group/task)	BrdU (1 × 200 mg/kg) for 3 days, 2 h pre-treatment [e1]; 2 h after 9×TMZ [e2]; 2 h before 10×TMZ [e3], or 13xTMZ [e4]	BrdU	↓BrdU compared to controls in e1–e4, and most reduced in e3 (*n* = 5–8/group)	/	/	/	/
Niibori et al., 2012 [123]	mouse/C57BL/6NTac x 129SvEvTac	M	6 weeks	TMZ (*n* = /) saline (DMSO, *n* = /)	25 mg/kg	12×; 1× on 3 consecutive days every week for 4 weeks	1 day	CFC	Compared to controls: ↓freezing ↓discriminaion in similar context; =freezing ↑ discrimination in dissimilar context (*n* = 11–12/group)	/	Ki67, NeuroD	↓Ki67, ↓NeuroD compared to controls (*n*= 4/group)	/	/		
Stone et al., 2011 [121]	mouse/ C57BL/6NTac × 129SvEvTac	M	8 weeks	TMZ (bilateral stimula-tion (BS) *n* = 24, non-stimulation (BNS) *n* = 17) saline (BS *n* = 24, BNS *n* = 17)	25 mg/kg	3×; 1× on day 1, 2, 3	7 weeks	MWM	=MWM escape latency during training across all groups, ↑MWM for S controls compared to other groups	BrdU (3 × 50 mg/kg/ day/8 h intervals for 3 consecutive days, either 3–5 days or 7 weeks postop (*n* = 5–8/group))	BrdU, BrdU-NeuN	3–5 days post-op: ↓BrdU and =BrdU-NeuN compared to controls; 7 weeks post-op: =BrdU =BrdU-NeuN for BNS and BS compared to controls (*n*= 5–8/group)	/	/	/	/
Garthe et al., 2009 [119]	mouse/C57BL/6J	F	6–8 weeks	TMZ (*n*= /) saline (*n*= /)	25 mg/kg	12×; 1× on 3 consecutive days every week for 4 weeks	2 weeks (MWM), 1 day pre-treatment, 1 day and 4 weeks post-treatment (rotarod, OF)	OF, rotarod, MWM	↓MWM compared to controls	BrdU (1 × 50 mg/kg, 4 days post-treatment)	BrdU	↓BrdU compared to controls	/	/	/	/

#### 3.2.4. ThioTEPA

ThioTEPA, another DNA alkylating agent with BBB permeability [136], is less commonly used in modern breast cancer treatment, and pre-clinical research into its effects is limited (Table 8). An early study using small samples of young adult mice investigated the dose-dependent impact of three daily injections of ThioTEPA (1, 5, or 10 mg/kg). BrdU labelling during the final treatment identified an immediate suppression of proliferating cells after just 2 h in response to all three dosages [92]. A follow up study assessed the long-term impact of three daily low dose (1 mg/kg) ThioTEPA treatments in mice using a cross-sectional time course assessment of cognition for up to 30 weeks [137]. BrdU treatment at the end of ThioTEPA treatment identified impaired survival of post-treatment generated neurons lasting 1–12 weeks following treatment. Neuronal survival deficits were accompanied by a modest but transient impairment on novel object and place recognition tasks 8–20 weeks following treatment. Given the time course for maturation and functional integration of adult-generated neurons [138], cognitive deficits emerging eight weeks post-treatment, a timepoint when new neurons normally would be functionally capable of integrating into new memory networks [139], it is likely that the reduced survival of new neurons contributed to the observed memory deficits. The performance of ThioTEPA-treated mice on the object and place recognition tasks was comparable to control levels after 30 weeks, suggesting cognitive recovery over time. In line with this observation, cell proliferation deficits were not persistent, with normal proliferation rates seen 30 weeks following the end of treatment [137]. This time profile indicates that ThioTEPA induces an early but likely temporary suppression of hippocampal neurogenesis following low dosage treatment. The long-term time course in response to higher dosages remains uninvestigated.

### 3.3. Mitotic Inhibitors

Mitotic inhibitors, while effective in inducing apoptosis through DNA damage, are especially toxic to proliferating non-malignant tissues, which may account for the high degree of neurotoxicity observed in response to these cancer treatments [140].

#### 3.3.1. Doxorubicin

Doxorubicin (DOX), an anthracycline drug, disrupts RNA production via DNA intercalation. Systemically administered DOX does not pass through the BBB [141,142], restricting it from directly entering the CNS. The neurogenic suppressing effects of DOX was first identified by Janelsins et al. [93]. Young adult mice were given a relatively high dosage (5 mg/kg) of three DOX treatments over the course of one week, and proliferating cells were labelled with BrdU one day following the final DOX injection. Brains collected 24 h following BrdU treatment revealed reduced cell proliferation in the SGZ in response to DOX, at levels comparable to similar treatments with BBB-permeable drugs including CPP and 5-FU. Christie and colleagues [17] found that a moderate dosage of DOX (2 mg/kg) spread out over four weekly injections was sufficient to impair both novel place recognition and 24 h context fear memory one week following the final DOX treatment in young adult male rats [17]. Cued fear memory remained unaffected, suggesting that the fear memory deficit is specific to hippocampally mediated memory processing. Relative to saline-treated controls, DCX expression was suppressed, indicating lower proliferation of hippocampal precursor cells and BrdU+ cells labelled one week following DOX treatment [17]. More recently, Park et al. [143] similarly found that four weekly treatments of 2 mg/kg DOX induced low rates of DCX and BrdU expression and impaired spatial memory in the water maze, confirming both reduced proliferation and one month survival of hippocampal cells generated following DOX treatment (Table 9).

Treatment with DOX has not consistently identified robust neurogenic depletion. Using the same DOX treatment protocol as Park et al. [143], Kitamura et al. [113] administered four weekly treatments of 2 mg/kg DOX and found impaired spatial location memory and place recognition one week following the final DOX treatment. Analyses of both cell proliferation and cell survival using BrdU treatments administered one week following the final DOX treatment or one day prior to beginning DOX treatment identified that the spatial memory deficit was accompanied by only modest, statistically non-significant reductions in cell proliferation and cell survival rates in the SGZ. This suggests that 2 mg/kg of DOX treatment, on its own, is not sufficient to drastically suppress neurogenesis in rats [113].

Treatment dosage and frequency also are important considerations in evaluating the neurotoxicity of anti-cancer drugs. In assessing the neurotoxic effects of a single treatment, Seigers et al. [73] found that one treatment of 5 or 10 mg/kg of DOX was not sufficient to impact rates of DCX or Ki-67 expression in male mice when assessed 3 or 16 weeks later. This suggests that a single dose of DOX, even at a high concentration, does not induce long-term neurotoxic effects on proliferating hippocampal cells.

#### 3.3.2. Paclitaxel

Paclitaxel (PTX; Taxol), a taxane agent used in breast and other cancer treatment, binds tubulin to hyper-stabilize microtubule polymerization, preventing normal axonal remodeling and transport and ultimately leading to cell death [144,145]. As with DOX, PTX is largely non-permeable to the BBB. A multi-treatment schedule of three 5 mg/kg treatments over the course of seven days was sufficient to suppress BrdU tagging of proliferating hippocampal cells 24 h following the final PTX treatment, indicating early inhibition of mitotic activity following systemic PTX treatment [93] (Table 10). Lower dosages of four PTX treatments (2 mg/kg) over the course of one week induced long-term neurogenic suppression including reductions in BrdU+ cells labelled three weeks following the final PTX treatment and reductions in Ki-67+ cells relative to controls [146].

The effects of PTX on hippocampal cell proliferation are mixed. A treatment dosage of 10 mg/kg of PTX administered daily over seven days (acute) or 15 PTX injections over 30 days (chronic) did not have any detectable effect on Ki-67+ progenitor cell proliferation or BrdU labelled cells immediately following the end of PTX treatment in male mice. Only a modest reduction in DCX+ cells was observed in both the acute and chronic PTX treatments, indicative of fewer neuroblasts relative to controls. Despite this modest reduction in neuroblasts, PTX-treated mice exhibited spatial learning impairments for platform location in the water maze [147].

Spatial memory deficits in the water maze were also observed following an intensive treatment regime of 12 systemic PTX injections of 20 mg/kg over the course of four weeks. Despite low BBB permeability, high PTX levels were detectable in blood serum and hippocampal tissue, although not in the cortex, within several hours of a single PTX injection, indicating that rapid uptake in the brain is selective to the hippocampus. BrdU tagging 24 h after the final PTX treatment identified fewer surviving BrdU+ and DCX+ cells in the DG relative to control mice when assessed three weeks later [148].

**Table 10 ijms-22-12697-t010:** Summary of pre-clinical studies investigating the effect of paclitaxel (PTX) treatment on hippocampal neurogenesis and memory processes.

Reference	Species/Strain	Sex	Age	Groups (*n*)	Dose	Treatment Schedule	Interval between Treatment and Tasks	Tasks	Chemo Behavior Memory	BrdU Schedule	NG Measures	NG Results	Intervention	Intervention Groups	Intervention Behavior	Intervention NG
Huehnchen et al., 2017 [148]	mouse/ C57BL/6J	M	9 weeks	PTX (*n* = 10–15) saline (cremophor + ethanol, *n* = 10–15)	20 mg/kg	12×; 1× every other weekday for 4 weeks	7 days	MWM, OF, EPM, FST, novelty suppression feeding, SP	↓ MWM compared to controls (*n* = 10–15/group)	BrdU (1 × 50 mg/kg 24 h post-treatment)	BrdU, BrdU-NeuN, DCX	↓BrdU ↓BrdU-DCX ↓BrdU-NeuN compared to controls (*n* = 12–14/group)	lithium (Li+)	170 μM administered before PTX (Saline/Li+; PTX/Li+)	↑MWM compared to PTX without Li+, =MWM for Li+ groups controls without Li+ (*n* = 11–13/group)	↑BrdU ↑BrdU-DCX ↑BrdU-NeuN compared to PTX without Li+, =BrdU =DCX =BrdU-NeuN compared to controls (*n* = 10–13/group)
Panoz-Brown et al., 2017 [146]	rat/Sprague Dawley	M	10 months	PTX (*n* = 6) saline (cremophor, *n* = 5)	2 mg/kg	4×; 1× every other weekday for 1 week	3 days pre-and 2 days post-treatment	Olfactory, SD, new and reverse learning	=olfactory =SD =new learning ↓reverse learning compared to controls	BrdU (1 × 100 mg/kg 18 days post-treatment)	BrdU, Ki67	↓BrdU ↓Ki67 compared to controls	/	/	/	/
Lee et al., 2017 [147]	mouse/ C57BL/6J	M	/	PTX acute and chronic (*n* = 11/group) Saline acute (*n* = 10) and chronic (*n* = 8)	10 mg/kg	3×; 1× every other weekday for 7 days (acute); or 15×; 1× every other weekday for 30 days (chronic)	3 days	MWM	↓ MWM compared to controls	BrdU (2 × 50 mg/kg/day either 1 (acute) or 4 (chronic) weeks post-treatment)	BrdU, Ki67, DCX	↓BrdU, ↓Ki67, ↓DCX for chronic compared to acute; =BrdU =Ki67 within groups	Zinc (Zn)	5 mg/kg/day for 8 days post-treatment (PTX + Zn, *n* = 8)	↑MWM compared to non-Zn groups, =MWM for PTX + Zn and controls	↑DCX for Zn compared to PTX, =BrdU, =Ki67 compared to other groups
Janelsins et al., 2010 [93]	mouse/ C57BL/6J	/	6–8 weeks	PTX (*n* = 6) saline (*n* = 8)	5 mg/kg	3×; 1× on days 1, 4, 7	/	/	/	BrdU (4 × 50 mg/kg/2 h intervals, 24 h post-treatment)	BrdU	↓BrdU compared to controls	/	/	/	/

#### 3.3.3. Docetaxel

Docetaxel (DTX), like paclitaxel, is a taxane agent that leads to microtubule dysfunction and cell death through the inhibition of microtubule dynamics [149,150]. A single systemic treatment of DTX (~30 mg/kg) did not produce reductions in Ki-67 expression after 3 or 16 weeks [73], but was sufficient to increase pro-inflammatory expression of TNF-α in the brain after one week [151], indicating an early inflammatory immune response. More extensive treatment over the course of three or four weeks was sufficient to impair novel object [152] and spatial novelty memory [153], indicating that overt hippocampal memory impairment may emerge gradually after cumulative treatments of DTX; however, the cumulative effects of the drug on hippocampal neurogenesis have not been investigated (Table 11).

### 3.4. Combination Therapies

Using a consistent approach to independently investigate multiple drugs is a powerful approach to identifying cognitive and physiological changes caused by unique chemotherapeutic agents with a high degree of control over dosage and timing of treatment and post-treatment assessments. However, many chemotherapy regimens used to treat common cancers, such as breast cancer, involve combination therapies of two or more separate agents, which may have an additive or interactive effect on cognitive function and neurotoxicity (Table 12).

#### 3.4.1. Methotrexate + 5-Fluorouracil

In their recent meta-analysis, Matsos and Johnston [21] identify MTX + 5-FU as among the most cytotoxic to the hippocampus and to memory processing. The very first systematic pre-clinical study of CICI in rodents used a combination of MTX + 5-FU, and identified both hippocampus-dependent memory impairments and frontal-lobe mediated disruptions of executive function [107]. Notably, this study and the series of studies that followed were among the few pre-clinical investigations of chemobrain to use female rodents, motivated by observations of cognitive disturbances specifically in breast cancer survivors, a predominantly female patient population. This initial behavioral study provided the first evidence for chemotherapy-mediated cognitive disturbances and led to an array of investigations into the cellular and physiological mechanisms underlying CICI following a range of drug treatments.

Following from their early behavioral work, Winocur and colleagues confirmed neurogenic depletion in response to combination therapy of MTX + 5-FU, drugs which are independently capable of producing several neurogenic and cognitive deficits. Female rats receiving three weekly injections of 37.5 mg/kg MTX and 50 mg/kg 5-FU exhibited spatial learning and memory deficits in the water maze as reflected in increased errors throughout training, as well as less precise search strategies during probe testing. In a non-matching-to-sample task (NMTS), a test of conditional rule learning, MTX + 5-FU-treated rats exhibited slow learning of the task over ten days, never achieving the level of performance of the controls [154,155,156]. When a 1–4 min delay between the sample trial and the subsequent test trial was implemented in the delayed version of the NMTS task (DNMTS), MTX + 5-FU-treated rats’ performance suffered, making far more errors than control rats in the subsequent test trial [154,155]. To successfully perform the DNMTS task, animals must engage both attentional and executive function processing mediated by the frontal lobe, as well as memory for the sample trial stimulus during the delay interval, which requires hippocampal processing. Both memory and attentional deficits are symptoms commonly reported by cancer patients following chemotherapy treatment, making these ideal tasks for capturing multiple domains of higher order cognitive functioning in response to chemotherapy treatment. When MTX + 5-FU treated rats were trained to perform a discrimination learning task in a T-maze, they acquired the task well. However, when presented with a competing interference task after acquiring the initial rule, MTX + 5-FU treated rats were subsequently impaired in re-learning the task [157].

BrdU labelling of dividing cells one day prior to sacrifice confirmed the MTX + 5-FU induced impairment in neuronal proliferation lasting more than one month following the end of chemotherapy treatment [155]. Following behavioral testing, MTX + 5-FU treated rats were found to have a 25% reduction in DCX levels in the SGZ persisting for over three months following the start of treatment [155,157,158]. These findings are consistent with studies using multiple dosages of MTX [75,76,77,78] or 5-FU alone [89,90,93,94,95,97], and identify the long temporal window of neurogenic suppression and the complex cognitive domains disrupted by MTX + 5-FU combination therapy in otherwise healthy female animals.

Both neurotoxicity and cognitive dysfunction induced by MTX + 5-FU treatments are exacerbated in the presence of breast cancer tumors. In a MMTVneu FVB oncogenic model of breast cancer, tumorigenesis begins around six months of age in transgenic mice [159]. MTX + 5-FU treatment after the onset of tumor development led to a suppression of DCX+ cells in the SGZ by 40% in both transgenic and wild-type mice [156], confirming earlier findings. Tumor bearing mice treated with saline did not exhibit any differences in rates of DCX+ expression in the SGZ relative to wild-type control mice, indicating that the presence of cancer on its own did not affect hippocampal cell proliferation rates [156]. Both saline and chemotherapy-treated tumorigenic mice did, however, exhibit a smaller volume of the hippocampus and fimbria/fornix as well as smaller overall brain volume, indicative of broader morphological changes in the brain in the presence of peripheral tumors. Parallel findings of reduced grey and white matter volume within the frontal lobe and hippocampus/parahippocampus region [160] and hippocampal deformity [161] have also been reported in breast cancer survivors for at least one year following treatment.

#### 3.4.2. Cyclophosphamide + Methotrexate + 5-Fluorouracil

Used in combination with 40 mg/kg CPP, 37.5 mg/kg MTX, and 75 mg/kg 5-FU also reduced cell proliferation in female rats [162]. Similar to findings using multiple treatments of MTX-5-FU [154,155,156,157] or CPP alone [17], four weekly injections of MTX-5-FU + CPP significantly reduced BrdU-labelled cell proliferation and impaired spatial memory performance during the water maze probe test one month following the end of treatment [162]. CPP + MTX + 5-FU treatment also induced high levels of histone deacetylase expression, indicative of histone modification, within the hippocampus and prefrontal cortex [162]. These findings point towards potential epigenetic changes mediating memory deficits induced by CPP + MTX + 5-FU.

#### 3.4.3. Doxorubicin + Cyclophosphamide

While various treatment regimes of DOX and CPP have been independently found to disrupt hippocampal neurogenesis, a similar pattern has been observed following combination treatment of these drugs. In male rats, a combination of 2 mg/kg DOX and 50 mg/kg CPP delivered weekly for four weeks impaired memory in the novel location recognition test. BrdU injections one week following the final treatment identified a significantly reduced number of proliferating cells in the DG at this time [113]. This same group later confirmed that a similar DOX + CPP treatment schedule impaired novel location memory and suppressed expression of Ki-67 one week following the final chemotherapy treatment [163]. As discussed earlier, a similar treatment protocol using CPP alone also suppressed BrdU+ cell proliferation in the DG; however, this same reduction was not observed in response to DOX alone [113]. These findings suggest that a moderate dosage of DOX on its own may be minimally neurotoxic to active hippocampal cell division. BrdU labelling of cells prior to combination chemotherapy treatment did not identify deficits in the survival of newly generated hippocampal neurons generated one day prior to treatment with either DOX or CPP; however, the combination therapy of both DOX + CPP resulted in a significant reduction in survival of pre-treatment BrdU labelled cells throughout the one month treatment and post-treatment interval [113].

Using the same DOX + CPP dosages and injection schedule as Kitamura and colleagues [113,163], Kang et al. [19] confirmed memory impairment in a novel object recognition task in young adult female mice. They also confirmed a reduction of both DCX expression and BrdU-labelled cells four weeks following the final chemotherapy treatment, indicating both reduced proliferation and cell survival, respectively. Mature granule cells exhibited a stunted morphology (reduced dendrite length and dendritic complexity) following DOX + CPP treatment. Reduced dendritic spine density was also observed in hippocampal sub-regions CA3 and CA1, suggesting that the neurotoxic effects of combination chemotherapy treatment of DOX + CPP propagate to impact existing hippocampal neurons downstream of the DG.

#### 3.4.4. Doxorubicin + Cyclophosphamide + 5-Fluorouracil

In ovariectomized female mice, when a high dosage of 4 mg/kg DOX and 80 mg/kg CPP was used in combination with 40 mg/kg 5-FU, two weekly treatments with the DOX + CPP + 5-FU cocktail was sufficient to impair spatial memory for the platform location during water maze probe testing almost three months following the end of chemotherapy treatment. BrdU labelling of proliferating cells during the first week of chemotherapy treatment identified reduced three-month survival of new neurons generated during the chemotherapy treatment period. A persistent reduction in new cell proliferation after three months was also confirmed by reduced Ki-67+ relative to controls [164]. These findings are likely representative of the long temporal profile of neurotoxic effects that are often reported in cancer patients receiving combination therapies of chemotherapy drugs.

**Table 12 ijms-22-12697-t012:** Summary of pre-clinical studies investigating the effect of combination treatments on hippocampal neurogenesis and memory processes.

Reference	Species/Strain	Sex	Age	Groups (*n*)	Dose	Treatment Schedule	Interval between Treatment and Tasks	Tasks	Chemo Behavior Memory	BrdU Schedule	NG Measures	NG Results	Intervention	Intervention Groups	Intervention Behavior	Intervention NG
Kang et al., 2018 [19]	mouse/ C57BL/6J	F	8 weeks	DOX + CPP (*n* = 5–17) saline (*n* = 5–17)	DOX 2 mg/kg + CPP 50 mg/kg	4×; 1/week for 4 weeks	10 days	TST, FST, NOR, OF	=NOR during training, ↓NOR 24 h after compared to controls (*n* = 17/group)	BrdU (1 × 100 mg/kg/day for 6 consecutive days starting 2 days post-treatment)	BrdU, DCX	↓ BrdU ↓DCX compared to controls 4 weeks after treatment (*n* = 5/group)	/	/	/	/
Jiang et al., 2018 [158]	rat/ Long Evans	F	12 weeks	5-FU + MTX (*n* = 13) saline (*n* = 10)	MTX 37.5 mg/kg + 5-FU 50 mg/kg	3×; 1× every 10 days for 30 days	10 days	SM, DL, NMTS	↓SM, ↓DL compared to controls	/	DCX	↓DCX compared to controls	PAN811	12 mg/kg 10 min after each treatment (PAN811 + 5-FU + MTX or saline + PAN811, *n* = 10–13/group)	↑SM ↑DL for PAN811 compared to 5-FU + MTX, =SM =DL compared to controls	↑DCX compared to MTX + 5-FU, =DCX compared to controls
Winocur et al., 2018 [156]	mouse/ MMTVneu FVB (exp) and FVB/NJ (controls)	F	9 months	Tumorigenic (Tg) + MTX + 5-FU, MTX + 5-FU (*n* = 10–18/group) Saline, saline + Tg (*n* = 10–018/group)	MTX 37.5 mg/kg + 5-FU 50 mg/kg	3×; 1/week for 3 weeks	Pre- (week 1–3) and post-treatment (week 8–12, 1 week)	MWM, CM, NMTS, CAL	[pre and posttreatment]↓MWM for Tg groups than non-Tg, ↓MWM for non-Tg chemo compared to controls (*n* = 12–18/group); =CM for all groups (*n* = /)	/	DCX	=DCX between controls with and without tumor; ↓DCX for Tg-chemo compared to all other groups (*n* = 11–18/group)	/	/	/	/
Winocur et al., 2016 [155]	rat/Long Evans	F	6 months	MTX + 5-FU in SE (*n* = 10) saline in SE (*n* = 10)	MTX 37.5 mg/kg + 5-FU 50 mg/kg	3×; 1/week for 3 weeks	7 days	MWM, CM, NMTS, DNMTS	↓MWM; =CM compared to controls	BrdU (2 × 100 mg/kg/8 h intervals administered 5 weeks post-treatment)	BrdU, DCX, BrdU-NeuN, BrdU-DCX	↓BrdU ↓DCX ↓BrdU-DCX ↓BrdU-NeuN compared to controls	EE	EE for 12 weeks pre-treatment (CHEMO-EE or controls-EE, *n* = 7–9/group).	↑MWM for EE than SE; =MWM for CHEMO-EE and controls-SE; =CM for all groups	↑BrdU ↑DCX ↑BrdU-NeuN ↑BrdU-DCX for chemo-EE than chemo-SE; ↑BrdU-DCX =BrdU-NeuN ↓DCX for saline-EE compared to saline-SE
Rendeiro et al., 2016 [164]	mouse/ C57BL/6J	F (ovariectomized)	12 weeks	DOX + CPP + 5-FU (*n* = 23) saline (*n* = 22)	DOX 4 mg/kg + CPP 80 mg/kg + 5-FU 40 mg/kg	2×; 1/week for 2 weeks	12 weeks	EPM, OF MWM, rotarod	↓MWM compared to controls	BrdU (1 × 50 mg/kg/day for 5 consecutive days starting after first chemo)	BrdU, Ki67, BrdU-NeuN	↓BrdU ↓BrdU-NeuN ↓Ki67 compared to controls when measured 13 weeks post-chemotherapy	FO [omega-3 (AIN-93G) + vitamin E + vitamin C]	0.16 g/kg DHA, 0.37 g/kg EPA, 0.05 g/kg DPA, 0.2 g/kg vitamin C, 0.185 g/kg vitamin E for 10 weeks starting 1- week post-treatment (CPP + DOX + 5FU + FO, FO, *n* = 22–23/group)	=MWM compared to groups without FO	=BrdU, =Ki67 =BrdU-NeuN for CHEMO with and without FO and between controls with and without FO when measured 13 weeks post-chemotherapy
Kitamura et al., 2015 [113]	rat/wistar	M	/	DOX + CPP (*n* = 6) saline (*n* = 6)	DOX 2 mg/kg + CPP 50 mg/kg	4×; 1/week; 4 weeks	SP 1 day before and 7 days post-treatment; other tasks 7 days post-treatment	light-dark test, NOL, SP	↓NOL compared to controls	BrdU (4 × 50 mg/kg/6 h intervals 7 d post-treatment and 24 h before last chemo	BrdU	↓BrdU for both cell survival and cell proliferation	/	/	/	/
Winocur et al., 2015 [157]	rat/Long Evans	F	5 months	MTX + 5-FU high or low interference (*n* = 8/group) saline-high or low interference (*n* = 7–9/group)	MTX 37.5 mg/kg + 5-FU 50 mg/kg	3×; 1/week for 3 weeks	DL training 1 week post-treatment and 1 week before interference, DL training 1 week after interference	DL	↓DL compared to controls and more so for high compared to low interference	/	DCX	↓DCX compared to controls	/	/	/	/
Winocur et al., 2014 [154]	rat/Long Evans	F	12 weeks	MTX + 5-FU (*n* = 10) saline (*n* = 9)	MTX 37.5 mg/kg + 5-FU 50 mg/kg	3×; 1/week for 3 weeks	7 days	MWM, CM, NMTS, DNMTS	↓MWM =CM compared to controls	/	DCX	↓DCX compared to controls	voluntary running	Running wheel in cage for 2 weeks pre-treatment (chemo-runners, control- runners (*n* = 9/group)	↑MWM for runners compared to non-runners and for chemo-runners compared to saline-runners; =CM for all groups	↑DCX for runners compared to non-runners, especially for chemo-runners
Briones & Wood, 2011 [162]	rat/Wistar	F	16 weeks	CPP + MTX + 5-FU (*n* = 12) saline (*n* = 12)	CPP 40 mg/kg + MTX 37.5 mg/kg + 5-FU 75 mg/kg	4×; 1/week for 4 weeks	14 days	MWM, DL	↓MWM ↓DL = after 3 days compared to controls	BrdU (1 × 100 mg/kg 24 days post-treatment/4 h prior to euthanasia)	BrdU	↓BrdU compared to controls	/	/	/	/

## 4. Interventions Protecting against Chemotherapy-Induced Neurogenic and Cognitive Impairment

Interventions aimed at preventing or minimizing cancer-treatment related neurotoxicity include behavioral, environmental, and pharmacological approaches. Evidence-based research in intervention and program development in various cancer survivors are in their early stages [165], with clinical trials largely limited to non-pharmacological pilot or proof-of-concept phases [166,167,168,169]. Here, we review the pre-clinical interventions shown to be effective in enhancing hippocampal neurogenesis and related cognitive performance following chemotherapy treatment.

### 4.1. Behavioral and Lifestyle Interventions

Physical exercise enhances proliferation of adult-generated newborn neurons in the hippocampus [170,171,172]. New neurons generated by, for example, running, functionally incorporate into new neuronal networks [139], contributing to the memory-enhancing effect of running. Running also improves memory performance [170], and upregulates neural plasticity-related proteins including neurotrophic factor BDNF [173] and CREB [174], a transcription factor that is critical for hippocampus-dependent memory [122,175]. Considerable overlap exists between the cellular mechanisms supporting running-enhanced cognition, and cellular mechanisms altered by chemotherapy treatment, including opposing effects on levels of neurogenesis, inflammatory cytokines, and plasticity-related proteins involved in memory formation, making running a promising environmental intervention that may protect against or prevent the loss of treatment-related hippocampal neural precursor cells. Fardell et al. [152] first identified the potential benefits of post-treatment running to ameliorate hippocampus-dependent memory deficits induced by combination chemotherapy of 5-FU + oxaliplatin (OX). Hippocampus-dependent context fear memory, novel object recognition, and spatial water maze deficits observed in 5-FU + OX treated mice were prevented when mice were given access to running wheels for one month following treatment.

Using combination therapy MTX + 5-FU, Winocur and colleagues [154] demonstrated that pre-posttreatment voluntary running prevented neurogenic depletion caused by chemotherapy in rats and also prevented hippocampus-dependent memory deficits in the water maze and DNMTS tasks, demonstrating for the first time that running is an effective behavioral intervention to protect against chemotherapy-induced neurotoxic effects on hippocampal cell proliferation and functional disruption of cognition. Park et al. [143] found that impaired spatial memory and reduced DCX and BrdU expression in DG were prevented in DOX-treated rats allowed access to one month of treadmill running [143]. These observations confirm both the overall neuroprotective benefits of an exercise intervention and the protective effects of post-treatment exercise in preventing chemotherapeutic neurotoxicity, in part by targeting a common hippocampal plasticity mechanism.

Voluntary running induces a range of neuroprotective effects, including enhancements in angiogenesis [176], hippocampal dendritic spine density and synaptogenesis [101,177], and BDNF expression [178,179], all of which may contribute to the observed neuroprotective effects following chemotherapy treatments. Both voluntary and involuntary exercise, such as treadmill running, has also been shown to confer protective effects on cognition and hippocampal neurogenesis [143], BDNF levels, mitochondrial function, and to reduce apoptosis and oxidative stress in the cortex [180] following DOX chemotherapy, although it is likely that stress associated with involuntary exercise may potentially limit the neuroprotective effects induced by running [181].

Environmental enrichment (EE) has been used as a non-invasive behavioral intervention to enhance hippocampus-dependent memory [182], enhance neurogenesis [183,184], promote dendritic arborization [185], and promote expression of the transcription factor CREB [186]. Winocur et al. [155] found that pre-treatment EE protected against the development of CICI. Rats reared for three months in the enriched environment (multi-level, tunnels, climbing ropes, toys, groups housed) and treated with a combination of MTX + 5-FU did not develop the deficits in the water maze or DNMTS tasks that were displayed by rats reared in a normal environment. Moreover chemotherapy-treated rats raised in the enriched environment exhibited rates of DCX+ and BrdU+ cells that were comparable to saline-treated controls. Walker et al. [187] found that social enrichment alleviates chemotherapy-related affective disturbances via oxytocin signaling [188]. Using a combination of DOX + CPP, they found that socially isolated mice displayed enhanced depressive-like behavior in the forced swim task which was accompanied by heightened pro-inflammatory cytokine expression of IL-6, IL-1β and TNF-α expression in the hippocampus [187].

Pre-clinical studies of exercise and EE have found that these interventions mediate cognitive and physiological processes that are negatively affected by chemotherapeutics. Both EE and running were independently found to prevent spatial learning and memory deficits following TMZ treatments in mice [120]. While hippocampal neurogenesis levels were not evaluated in this study, it is likely that the neurogenic-boosting effects of both EE and running compensate for a TMZ-induced neurogenic suppression [119], and presumably mediated the observed memory preservation [120].

Voluntary running and EE have been shown to independently enhance different components of the neurogenic process, with running promoting cell proliferation and EE enhancing survival of adult-generated hippocampal cells [172]. As is well established, enhanced proliferation and cell survival are protective against chemotherapy-induced hippocampal dysfunction and associated cognitive deficits.

Although behavioral and lifestyle interventions are promising non-invasive clinical options, it has yet to be shown that these interventions are reliably successful in preventing CICI in cancer survivors [189]. Pain and fatigue, as well as psychosocial factors such as perceived stress and social support, likely confound the potential beneficial effects of running (see [190] for a recent review of the efficacy of exercise interventions in breast cancer survivors). While findings in adult patients are mixed, a post-treatment exercise intervention study in pediatric brain tumor survivors found that 12 weeks of group exercise significantly increased hippocampal volume of patients who had previously received both cranial radiation and chemotherapy treatments when compared to their pre-exercise levels [191]. While the aerobic intervention was not associated with enhanced memory processing, it did improve processing speed, a cognitive function commonly impaired by chemotherapy treatments. Notably, this robust enhancement of hippocampal volume was observed in response to group exercise, but not seen when patients performed the same activities individually. The physiological changes in the hippocampus induced by of aerobic exercise combined with the environmental enrichment provided by a group exercise activity may have interacted to potentiate the neuroprotective effects of the exercise intervention in this patient population. A similar potentiating effect has been observed in rodents allowed to engage in voluntary running in groups relative to those running in isolation [192].

### 4.2. Pharmacological Interventions

#### 4.2.1. Fluoxetine

Antidepressant selective serotonin reuptake inhibitors (SSRI) increase both the proliferation and survival of post-natally generated hippocampal neurons [193]. The SSRI fluoxetine is capable of boosting rates of hippocampal neurogenesis over basal levels [194], and has been effective in preventing disease-related neurogenic impairments in a range of clinical disorders [195,196,197,198]. In a series of studies independently assessing MTX and 5-FU, Wigmore and colleagues have shown that fluoxetine, given prior to and throughout the chemotherapy treatment period, is effective in preventing chemotherapy-induced neurotoxicity and memory impairments. Preventative delivery of fluoxetine through drinking water prior to or throughout 5-FU treatment was protective against the loss of both pretreatment-generated hippocampal neurons and post-treatment proliferation in the SGZ in adult male rats [94,95]. A similar fluoxetine supplement following 5-FU treatment was not sufficient to recover hippocampal cell loss induced by 5-FU [94] or to prevent spatial memory deficits in the novel location recognition task, suggesting that this pharmacological intervention may be best as a preventative treatment.

Oral administration of fluoxetine beginning one week prior, throughout, and following MTX treatment is also effective in preventing MTX-induced neurogenic impairments in BrdU+ cell survival during the MTX treatment period, and in protecting against post-treatment impairments in cell proliferation (both Ki-67 and DCX+ cells) for at least one month post-MTX treatment in male rats [75]. Preventative administration of fluoxetine prior to MTX treatment and persistent fluoxetine treatment throughout the pre- and post-MTX treatment time period were equally effective in protecting against chemotherapy-related neurogenic cell survival, and proliferation [78].

#### 4.2.2. Lithium

Lithium, a mood stabilizer used in depression and bipolar therapy, is thought to promote cognition and neurogenesis rates though GSK-3B inhibition [199]. Lithium has been used as an effective intervention in several disorders in which neurogenic depletion and cognitive impairment are typically observed, including an Alzheimer’s mouse model [200,201], Down’s Syndrome model [202], and in response to cranial irradiation [203]; however, it has not been extensively tested as an intervention for CICI. In their study of PTX, Huehnchen and colleagues [148] found that both spatial memory impairment and neurogenic reduction following a dense-dosage PTX treatment was prevented by injection of lithium carbonate (Li+) prior to PTX treatment, indicating this as a promising preventative treatment. Li+ treatment on its own was not sufficient to enhance memory performance or neurogenic rates over control levels, though previous investigations have found it to be an effective neurogenic booster following chronic administration over four weeks [204].

#### 4.2.3. Metformin

Metformin, an AMPK agonist, is used for glucose regulation in type 2 diabetes [205]. Metformin has been effective in boosting hippocampal neurogenesis and spatial memory in healthy rodents via activation of an atypical PKC-CBP pathway, promoting differentiation of neuronal precursor cells [206]. When used preventatively prior to and throughout MTX treatment in rats, metformin was effective in preventing memory deficits in novel location recognition and novel object recognition tasks, as well as in protecting against the suppression of Ki-67+ and DCX+ cells in the SGZ observed in response to MTX treatment alone [78].

#### 4.2.4. Donepezil

Donepezil, a cholinesterase inhibitor and NMDA receptor antagonist, has been effectively used to reduce cognitive impairment in mild-to-moderate Alzheimer’s disease [207]. In the presence of cholinergic inhibition, donepezil treatment is sufficient to rescue neurogenic deficits in rats through enhancement of phosphorylated CREB and BDNF [208]. The effectiveness of donepezil in rescuing neurogenic depletion in response to chemotherapy treatments has not yet been evaluated, but is a promising intervention, as systemic donepezil given during MTX + 5-FU treatment attenuated memory deficits in the DNMTS task in rats [209].

#### 4.2.5. Melatonin

Melatonin is an endogenous hormone involved in circadian regulation and has antioxidant and anti-apoptotic properties [210]. Melatonin has been found to have an oncostatic effect in several types of cancers [211,212], and has shown promise in protecting against neurogenic loss in several neurological disorders [213,214,215], in normal aging [216], and in response to irradiation [217]. Two weeks of melatonin treatment given prior to and during the MTX treatment period prevented neurogenic and memory impairments in novel object and novel location recognition experienced by rats treated with MTX alone. Further, initiating the two weeks of melatonin treatment during MTX treatment prevented the development of neurogenic and cognitive impairment. No cumulative benefits on cognition or neurogenesis rates were evident in rats administered melatonin for four weeks throughout the pre- and post-MTX treatment period [76]. Following the same approach, melatonin was equally effective in both preventing and rescuing neurogenic deficits and spatial memory deficits in novel location recognition following an extensive five-treatment 5-FU protocol [97]. As with MTX, no cumulative benefit was evident when melatonin was administered throughout both the pre- and post-5-FU treatment period, suggesting that melatonin supplements are equally effective at preventing memory deficits and neurotoxicity induced by MTX or 5-FU when proactively administered prior to or following the initiation of chemotherapy treatment. Melatonin treatment also prevented the increases in p21 expressing cells within the SGZ seen in rats treated with 5-FU alone, indicating reduced cell cycle arrest within the neurogenic zone [98].

#### 4.2.6. Zinc

Systemic zinc supplements are capable of enhancing hippocampal cell proliferation and zinc transporter levels in the pre-synaptic mossy fiber terminals in the DG of rats [218]. In their investigation of PTX-induced neuronal effects in the hippocampus, Lee and colleagues [147] found that PTX induced depletion of zinc transporters and vesicular zinc in the pre-synaptic mossy fiber terminals of DG neurons, and induced a modest reduction in proliferated neuroblasts in the DG. A zinc supplement was effective in promoting neuroblast differentiation of DCX+ cells, and preventing the spatial learning deficits in the water maze task observed in mice treated with PTX [147].

#### 4.2.7. Curcumin

Curcumin, a polyphenol compound and natural plant extract with anti-inflammatory and anti-oxidant properties [219] has been used as an intervention in a range of neurodegenerative and psychiatric disorders and in enhancing memory and hippocampal neurogenesis during normal aging [220,221,222]. Curcumin has been studied for its anti-tumor properties in many non-CNS cancers [223,224], and the cellular mechanisms promoting tumor suppression may similarly protect against the neurotoxic properties of certain chemotherapy drugs. In one study, cisplatin reduced DCX expression levels within the SGZ, and led to a reduction in hippocampal dendritic spine density. Both neurogenic reductions, dendritic synaptogenesis, and spatial and object memory impairments induced by cisplatin treatment were prevented by curcumin administration in mice [105].

#### 4.2.8. Hesperidin

The anti-oxidant hesperidin is a citrus flavonoid which promotes synaptic plasticity in part through activation of the ERK/PKA/CREB pathway [225]. In rats administered hesperidin following a single MTX treatment, the memory deficits in novel object and location recognition tasks and hippocampal neurogenic deficits in BrdU, DCX, and Ki-67 expression observed in rats treated with MTX alone were prevented [77]. MTX-induced increases in cell-cycle arrest marker p21 in the hippocampus were also alleviated by post-treatment hesperidin administration [79]. The effectiveness of hesperidin in promoting hippocampal cell proliferation and preventing cell death is somewhat surprising given that hesperidin has been explored as an anti-cancer treatment due to its pro-apoptotic and anti-proliferative actions in some cell types through p53 accumulation [226].

#### 4.2.9. Ginsenoside Compound K

Ginsenoside Compound K, a major metabolite of ginsenoside Rb1, has been shown to suppress tumor growth and enhance apoptosis of tumor cells via enhanced p53 signaling [227], and to protect against inflammation and cognitive dysfunction by stimulating LXR-alpha [228,229]. Co-administration of Ginsenoside Compound K with CPP treatment minimized the loss of BrdU and DCX-expressing cells in a dose-dependent manner, with higher dosages of up to 10 mg/kg almost completely protecting against neurogenic loss in response CPP treatment relative to controls [112].

#### 4.2.10. IGF-1

Insulin growth factor-1 (IGF-1), an endogenous growth factor [230], has previously been used to protect against neurogenic depletion in aging [231]. Given the overlapping mechanisms disrupted in response to chemotherapy treatment and aging [51], and the link between chemotherapy treatment and biomarkers indicative of accelerated aging [232,233], the use of IGF-1 is a promising candidate intervention to protect against CICI. Co-administration of IGF-1 and a high dosage of CPP largely prevented the loss of BrdU labelled hippocampal neurons observed in response to CPP treatment alone [93].

### 4.3. Interventions Requiring Further Investigation of Their Effects on Neurogenesis

Additional studies are in the early stages of determining the effectiveness of certain pharmacological interventions (examples include the anti-oxidant D-methionine [103], nicotine supplement cotinine [163], and p53 mediator pifithrin (PFT)-μ [104]) for preventing neurogenic depletion when given in combination with traditional chemotherapies (see intervention section of Tables). A variety of other pharmacological interventions have been used in combination with various chemotherapeutic drugs and have shown promise in reducing neurotoxicity, including reducing the expression of pro-inflammatory cytokines using Aspirin [234], the ginseng-derived compound Ginsenoside RG1 [235], and the mango extract Mulmina [236].

## 5. Limitations and Considerations

As noted in a recent review by Matsos and Johnston [21], despite the control over confounding variables afforded by pre-clinical rodent studies of chemobrain, methodological heterogeneity between studies complicates the understanding of cognitive and physiological dysfunction following chemotherapy treatments. Inconsistencies in treatment regimes (drug combinations, dosages, frequency, schedule) as well as differences in species and strain, age at the time of treatment, limited sample sizes, variable intervals in treatment, behavioral testing, and brain tissue collection, behavioral test protocols, cross sectional vs. longitudinal designs, and neurogenic labelling and assessment methods are all among the factors that limit comparisons across studies of the various chemotherapeutic drugs. To address this issue, the *International Cognition and Cancer Task Force* has published recommendations in order to standardize test batteries and protocols across pre-clinical studies [237] which, if followed, should help to standardize comparisons across studies.

Most studies, particularly those assessing behavioral and pharmacological interventions, have used exclusively male mice or rats, which raises an important issue for understanding the effectiveness and the consequences of these treatments for females. Given the focus on many of the reviewed chemotherapeutic drugs and their use in breast cancer treatment in women, it is critical to expand these pre-clinical studies to include the use of female as well as male mice to determine how these drugs may differentially affect the brain, cognition, and affective behavior in both sexes.

In this review, we focused on neurogenic disruptions in cell proliferation and cell survival in response to chemotherapy treatment. It is important to note that other pathologies, including microtubule dysfunction, reduced dendritic branching and spine development [51,238], suppressed gliogenesis and mature and precursor oligodendrocytes [89,101], reduced myelination and related white matter degradation [239], and increased rates of apoptosis [89] likely contribute to hippocampal neurotoxicity, reduced hippocampal volumes, and hippocampally-mediated cognitive impairment.

### Problems with Comparing the Neurotoxic Profiles of Drugs

As noted, a substantial problem with many pre-clinical studies of chemotherapy-induced neurotoxicity lies in the variability of study protocols. To address methodological inconsistencies, several studies have used a systematic approach to comparing the cognitive and neurotoxic effects of particular chemotherapy drugs by concurrently examining individual drugs using the same treatment schedule and neurogenic labelling and staining protocols. Janelsins et al. [93] independently investigated the BBB-permeable drugs CPP and 5-FU along with BBB non-permeable DOX and PTX and identified consistent neurogenic reductions relative to saline controls for each drug, despite their differences in BBB permeability. Co-staining of hippocampal tissue with cleaved caspace-3 confirmed that lower BrdU expression was indicative of lower proliferation rates 24 h following the final chemo treatment, and not due to enhanced apoptosis relative to saline-treated controls [93]. Similarly, Christie et al. [17] concurrently administered CPP and DOX in independent groups of rats using a multi-injection schedule over four weeks and found converging evidence for a 50–90% loss of both DCX and BrdU-labelled cells in the hippocampus. Of the remaining DCX-expressing cells in both chemotherapy treatment groups, they identified morphological abnormalities of the dendritic processes and ectopic migration of cells from the SGZ into the hilus, suggesting that newly generated cells will not normally develop and functionally incorporate into the granule cell network, thereby contributing to further functional impairment, and impairment of hippocampus-dependent memory.

In a comprehensive study, Seigers and colleagues [73] directly compared neurogenesis rates in young adult male mice in response to six separate chemotherapy drugs. Mice were given a single treatment with either CPP, docetaxel, DOX (high or low dose), 5-FU, MTX (high or low dose), or topotecan and sacrificed either 3 or 16 weeks later to assess the short and long-term treatment effects on neurogenesis. Their findings suggest that the neurogenic depleting effects of hippocampal neurogenesis may be limited to a more immediate post-treatment window, as no notable differences were observed in Ki-67 or DCX expression following any of the chemotherapy treatments at either the 3 or the 16 week post-treatment intervals Their results further suggest that hippocampal neuronal precursor development may return to basal rates in the weeks following treatment. Alternatively, unlike studies using multiple treatment timepoints, a single treatment of any one chemotherapeutic drug may not be sufficient to induce long-lasting neurotoxic effects on hippocampal neurogenesis [17,93].

## 6. Translational Interventions and Patient Applications

As measuring neurogenesis levels in vivo is not possible in humans, measures of hippocampal volume and medial-temporal lobe cortical thickness are used as a proxy for hippocampal integrity. In line with findings of reduced hippocampal neurogenesis in pre-clinical models, various chemotherapy treatments are associated with reduced hippocampal and parahippocampal volume in patients [160,161,240], correlating with impaired memory performance up to a year following treatment [160]. Other imaging studies, however, have failed to find a difference in hippocampal volume following chemotherapy [241], suggesting that gross morphological changes in response to treatment may be difficult to detect given the relatively low spatial resolution of most MRI studies. The prevalence of other mechanistic disturbances, including dendritic and spine changes, white matter changes, and neuroinflammation, among others, contribute to potential differences in gross hippocampal morphology detectable by in vivo neuroimaging techniques in humans and confound interpretations of disruptions localized to the hippocampal subregions that may be indicative of neurogenic impairment. Postmortem analyses have confirmed reduced hippocampal DCX levels following systemic chemotherapy treatment, in combination with radiation [20].

Understanding the temporal pattern of cancer-related cognitive impairment is important. Longitudinal tracking of breast cancer survivors has revealed cognitive or neural abnormalities in 40% of patients prior to treatment, caused by disease onset and likely exacerbated by psychosocial factors [242,243,244,245]. Blommaert et al. [241] report gray matter reductions in breast cancer patients following resection of cancerous tumors even in the absence of chemotherapy treatment, indicative of cortical atrophy associated with the cancer alone. Cognitive disturbances are evident in up to 75% of patients during and following treatment [242]. Many pre-clinical studies have been limited to cross-sectional studies of post-treatment behavioral and physiological effects relative to control groups. While important, this approach lacks the ability to track the temporal profile of cognitive and neurotoxic changes in the brain following treatment in order to better characterize the long-term efficacy of CICI interventions. Further, the effectiveness of behavioral and pharmacological interventions in pre-clinical studies does not always directly translate to similar savings when used in cancer patients [168], highlighting the need for further investigation into the efficacy of these interventions in cancer survivors and consideration of the environmental and sociocultural risk factors which may predict the severity of chemobrain and the responsiveness of a given intervention in lessening cognitive impairment in cancer survivors.

## Figures and Tables

**Figure 1 ijms-22-12697-f001:**
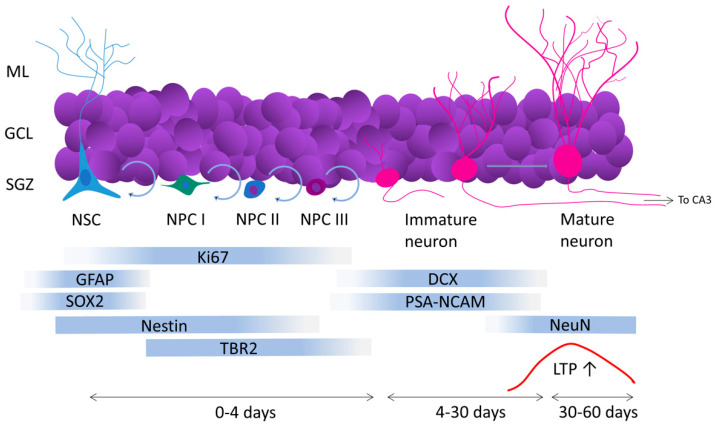
Schematic of the adult hippocampal neurogenesis process. Abbreviations: DG, dentate gyrus; ML, molecular layer; GCL, granular cell layer; SGZ, subgranular zone; GFAP, glial fibrillary acidic protein; DCX, doublecortin; TBR2, T-box brain protein 2; PSA-NCAM, polysialylated neural cell adhesion molecule.

**Table 4 ijms-22-12697-t004:** Summary of pre-clinical studies investigating the effect of cytarabine treatment on hippocampal neurogenesis and memory processes.

Reference	Species/Strain	Sex	Age	Groups (*n*)	Dose	Treatment Schedule	Interval between Treatment and Tasks	Tasks	Chemo Behavior Memory	BrdU Schedule	NG Measures	NG Results	Intervention	Intervention Groups	Intervention Behavior	Intervention NG
Dietrich et al., 2006 [101]	mouse/CBA	/	6–8 weeks	Cytarabine (*n* = 5) saline (*n* = 5)	250 mg/kg	3×; 1× on days 1, 3, 5	/	/	/	BrdU (1 × 50 mg/kg administered 4 h before perfusion)	BrdU, BrdU-DCX	↓BrdU starting at 7 days and most reduced 56 days post-treatment; ↓BrdU-DCX starting 1 day but = 56 days post-treatment	/	/	/	/

**Table 6 ijms-22-12697-t006:** Summary of pre-clinical studies investigating the effect of carmustin treatment on hippocampal neurogenesis and memory processes.

Reference	Species/Strain	Sex	Age	Groups (*n*)	Dose	Treatment Schedule	Interval between Treatment and Tasks	Tasks	Chemo Behavior Memory	BrdU Schedule	NG Measures	NG Results	Intervention	Intervention Groups	Intervention Behavior	Intervention NG
Dietrich et al., 2006 [101]	mouse/ CBA	/	6–8 weeks	carmustine (BCNU) (*n* = 5) saline (*n* = 5)	10 mg/kg	3×; 1× on days 1, 3, 5	/	/	/	BrdU (1 × 50 mg/kg); 4 h before perfusion	BrdU, BrdU-DCX	↓BrdU 1 and 42 days; =BrdU-DCX 1-day post-treatment compared to controls	/	/	/	/

**Table 8 ijms-22-12697-t008:** Summary of pre-clinical studies investigating the effect of thioTEPA treatment on hippocampal neurogenesis and memory processes.

Reference	Species/Strain	Sex	Age	Groups (*n*)	Dose	Treatment Schedule	Interval between Treatment and Tasks	Tasks	Chemo Behavior Memory	BrdU Schedule	NG Measures	NG Results	Intervention	Intervention Groups	Intervention Behavior	Intervention NG
Mondie et al., 2010 [137]	mouse/C57BL/6J	M	8–9 weeks	thioTEPA (*n* = 4–10) saline (*n* = 4–10)	10 mg/kg	3×; 1× on 3 consecutive days	[e2] 2, 4, 8, 12, 20, or 30 weeks	[e2] FST, TST, NOR, NOL	[e2] Compared to controls: =NOR 1- and 2-weeks, ↓NOR 8- and 12-week post-treatment. =NOL 2–8 weeks, ↓NOL 20 weeks post-treatment (*n* = 8–10/group)	[e1] BrdU (1 × 50 mg/kg immediately after treatment; or 30 min prior to perfusion)	[e1] BrdU	[e1] ↓BrdU from 4, 8, and 12 weeks compared to controls, =BrdU at 30 weeks (proliferation); ↓cell survival within 1 week of chemo compared to controls (*n* = 4/group)	/	/	/	/
Mignone et al., 2006 [92]	mouse/C57BL/6J	/	6 weeks	thioTEPA (*n* = 3) saline (*n* = 3)	1, 5, or 10 mg/kg	3×; 1× on 3 consecutive days	/	/	/	BrdU (1 × 200 mg/kg with final treatment)	BrdU	Dose-dependent ↓BrdU compared to controls	/	/	/	/

**Table 9 ijms-22-12697-t009:** Summary of pre-clinical studies investigating the effect of doxorubicin (DOX) treatment on hippocampal neurogenesis and memory processes.

Reference	Species/Strain	Sex	Age	Groups (*n*)	Dose	Treatment Schedule	Interval between Treatment and Tasks	Tasks	Chemo Behavior Memory	BrdU Schedule	NG Measures	NG Results	Intervention	Intervention Groups	Intervention Behavior	Intervention NG
Park et al., 2018 [143]	rat/Wistar	M	6 weeks	DOX (*n* = 15) saline (*n* = 15)	2 mg/kg	4×; 1/week for 4 weeks	/	MWM, step down avoidance task	↓avoidance ↓MWM compared to controls	BrdU (100 mg/kg/day for 7 days following the first week of exercise)	DCX, BrdU-NeuN	↓BrdU-NeuN ↓DCX compared to controls	Exercise (low intensity treadmill)	40 min of exercise/day for 6 days/week for 4 weeks (DOX + exercise, saline + exercise, *n* = 15/group)	↑MWM ↑avoidance task compared to DOX without exercise, = avoidance task between controls with and without exercise	↑DCX ↑BrdU-NeuN compared to without exercise, ↑DCX ↑BrdU-NeuN for saline + exercise compared to DOX + exercise
Seigers et al., 2016 [73]	mouse/ C57BL/6J	M	11 weeks	DOX (*n* = /) saline (*n* = /)	5 or 10 mg/kg	1×	/	/	/	/	DCX, Ki67	=DCX =Ki67 compared to controls when sacrificed 3- and 16-weeks post-treatment	/	/	/	/
Kitamura et al., 2015 [113]	rat/Wistar	M	/	DOX (*n* = 6) saline (*n* = 6)	2 mg/kg	4×; 1/week for 4 weeks	SP 1 day before and 7 days post-treatment; other tasks 7 days post-treatment	light-dark test, NOL, SP	↓ NOL recognition compared to controls	BrdU (4 × 50 mg/kg/6 h intervals) 7 d post-treatment and 24 h before last treatment	BrdU	NS ↓BrdU for cell survival and proliferation	/	/	/	/
Christie et al., 2012 [17]	rat/Athymic Nude	M	8 weeks	DOX (*n* = 9) saline (*n* = 8)	2 mg/kg	4×; 1/week for 4 weeks	7 days	NOL, CFC	↑NOL familiarization, NS ↓NOL choice trial, ↓ CFC compared to controls	BrdU (1 × 100 mg/kg/day for 6 days, starting 2 days post-treatment)	BrdU, BrdU-NeuN, DCX	↓BrdU-NeuN, =BrdU, ↓DCX compared to controls	/	/	/	/
Janelsins et al., 2010 [93]	mouse/ C57BL/6J	/	6–8 weeks	DOX (*n* = 6) saline (*n* = 8)	5 mg/kg	3×; 1× on days 1, 4, 7	/	/	/	BrdU (4 × 50 mg/kg/2 h intervals, 24 h post-treatment)	BrdU	↓BrdU	/	/	/	/

**Table 11 ijms-22-12697-t011:** Summary of pre-clinical studies investigating the effect of docetaxel (DTX) treatment on hippocampal neurogenesis and memory processes.

Reference	Species/Strain	Sex	Age	Groups (*n*)	Dose	Treatment Schedule	Interval between Treatment and Tasks	Tasks	Chemo Behavior Memory	BrdU Schedule	NG Measures	NG Results	Intervention	Intervention Groups	Intervention Behavior	Intervention NG
Seigers et al., 2016 [73]	mouse/ C57BL/6J	M	11 weeks	DTX (*n* = /) saline (*n* = /)	22 mg/kg	1×	/	/	/	/	DCX, Ki67	=DCX =Ki67 compared to controls when sacrificed 3- and 16-weeks post-treatment	/	/	/	/

## Data Availability

Not applicable.

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
