# Peer review of "Chemotherapy-Induced Cognitive Impairment and Hippocampal Neurogenesis: A Review of Physiological Mechanisms and Interventions"

_ijms, 2021, doi:10.3390/ijms222312697_

Round 1
Reviewer 1 Report
The manuscript is related to chemotherapy-induced cognitive impairment associated with adult hippocampal neurogenesis. The intent of authors was to “review the pre-clinical rodent literature to identify how various chemotherapeutic drugs affect hippocampal neurogenesis and induce cognitive impairment”. This aim is in line with the title of the manuscript. However, the authors describe also “factors such as physical exercise and environmental stimulation that may protect against chemotherapy-induced neurogenic suppression and hippocampal neurotoxicity” and “pharmacological interventions that target the hippocampus and are designed to prevent or reduce cognitive and neurotoxic side effects of chemotherapy”. The title does not summarize the other two topics. Therefore, it should be changed. Moreover, there are also some major issues that need to be addressed:
In general, preclinical studies of the effects of anticancer drugs are being conducted in laboratory animals in order to find possible mechanisms by which drug treatment induces changes at the cellular level. This review paper is focused on the effects of chemotherapeutic drugs from 3 different classes. First, it is not clear what criteria were chosen for considering the selected drug. Among several antimetabolites drugs to be used against cancer, only a few (Methotrexate, 5_Fluorouracil, Cisplatin, Cytarabine) were considered or of the 5 known categories of alkylating agents, only four (Cyclophosphamide, Carmustin, Temozolomide, ThioTEPA) were included in the paper. This also applies to mitotic inhibitors. I wonder whether the authors can justify their choice. Second, are there any differences in behavior and neurogenic reduction between mice and rat treated with the same chemotherapeutic drug?
Also, I believe that each subsection should be followed by a brief discussion that would give some insight into how and why different drugs are used.
The tables after each section are not legible, they take up a lot of space and contain little information.
Line 920: Replace title “Compound K” with “Ginsenoside Compound K”
Line 921: “Compound K is a ginseng metabolite” This sentence needs clarification.
Line 929-931: Remove this sentence “Various growth factors such as BDNF, VEGF, and IGF are necessary for postnatal neurogenic growth and enhancing growth factor expression may promote hippocampal neurogenesis [199–201].”
Thorughout the text, replace the word “watermaze” with “water maze”.
Reviewer 2 Report
In general, the authors did a systematic and exhaustive work in reviewing the available literature. The topic of the review is timely and conceptually challenging, but they did an extraordinary job in gathering all the information.
I have a few main concerns and several minor suggestions to further improve the review.
Main concerns:
- One of the main problems of this review paper is the Introduction section. The systematic approach of the literature review drastically contrast with the vague character of the first paragraphs. The absence of an introductory section describing adult neurogenic niches, stages, and the whole process can be problematic for non-expert reviewers.
- Second, there is one study on human AHN and cancer patients that has not been included in this review. Please include Monje et al., Annals of Neurology, 2007 in this review. This is a seminal study that examined and described AHN impairments in cancer patients. The results of this study should be discussed in detail.
- A graphical scheme of the AHN stages on which distinct chemotherapy drugs act preferentially would help the readers to have a general overview of these effects.
Minor points:
- In the Abstract, the sentence “These 17 findings suggest that systemically administered chemotherapy drugs are capable of mechanistically inducing neurotoxicity and behavioral disturbances” is repetitive and does not add any substantial information to the Abstract.
- In lines 84-89, the sentences “Rates of adult-generated granule cell proliferation and differentiation can be measured using transiently expressed endogenous markers of mitotic division of hippocampal neural precursor cells and immature neurons (Ki-67, and doublecortin, DCX). These markers are expressed in neural precursors for approximately 2-3 weeks following division [32–34], providing a window into proliferative activity occurring within the weeks prior to sacrifice” are not accurate. Ki67 is expressed only during proliferation, whereas DCX is expressed during several weeks. The authors should clarify what they mean with these sentences and indicate at which precise time is expressed each of the markers mentioned.
- In lines 89-95, the authors state that “The synthetic ligand 5-bromo-2'-deoxyuridine (BrdU) allows for more precise labelling of actively dividing neurons for an acute period following BrdU injection, with persistent expression. Unlike Ki-67 and DCX, BrdU will only be expressed briefly in cells following treatment, and therefore provide a ‘snapshot’ of cell mitotic activity at the time of labelling. BrdU expression levels post-mortem will depend largely upon the BrdU treatment schedule (single injection, or repeated daily injection) prior to sacrifice, as well as the treatment-sacrifice interval [35]”. This is an important conceptual error. BrdU is not expressed by the cells. BrdU is injected, and cells under division at that precise time uptake it. Afterwards, BrdU incorporation can be detected, but cells do not express BrdU. They also made important conceptual errors in terms of the acute/long-term periods in those sentences. They should revise these concepts and re-write this paragraph. Including a general scheme of adult neurogenesis and indicating where BrdU is uptaken could help understanding these data to non-expert readers. A more detailed description on how differentiation/maturation can be studied using BrdU may be useful too.
- The purpose of including several sentences (i.e., those included between lines 96 and 103) is unclear. What to these lines add to the rationale of this review? In this and the next paragraph, the authors only indicate how complicate is the disentangling of the available literature. However, that should be the aim of writing a review about a topic. In fact, they systematically review all the available literature and the following sections are easy to understand.
Reviewer 3 Report
The review entitled “Chemotherapy-induced Cognitive Impairment and Hippocampal Neurogenesis” deals with a very important issue in the context of cancer and it is related, in particular, to the cognitive alterations suffered by subjects that have been sujected to chemotherapy. This evidence known as “Chemobrain” was first related mainly to physcological distress, nevertheless, the advances in the research field, also in neuroimaging techniques, have brought knowledge to light that this chemotherapy-induced cognitive impairment (CICI) is also related to brain neurotoxicity alterations including brain volume and white and grey matter integrity.
The information included in the review represents a significant and detailed contribution in the field of adverse events of chemotherapy in the brain. Moreover, the manuscript is well organized, written and comprehensibily described despite the high amount of information included.
I have just some comments/suggestions:
- Tables sommarizing described information are very useful, but they contain such as much information that because of space limits it is difficult to follow them. It would be of help to include lines into the table.
- Page 4, lines 172-173, two groups of interval references have been included in separated lines, is this corret?
Round 2
Reviewer 1 Report
The authors have improved on the manuscript.
Reviewer 3 Report
Authors have considered all suggestions so that much improving the manuscript.
The new figure 1 is of great help in understanding the development of new neurons. There is an abbreviation missing, TBR2, which meaning is not specified anywhere through all the text, nor in the supplementary file.